**DOI: 10.1038/ncomms16047**　　**OPEN**

# The ancestral flower of angiosperms and its early diversification

Hervé Sauquet[1], Maria von Balthazar[2], Susana Magallón[3], James A. Doyle[4], Peter K. Endress[5], Emily J. Bailes[6], Erica Barroso de Morais[5], Kester Bull-Hereñu[7], Laetitia Carrive[1], Marion Chartier[2], Guillaume Chomicki[8], Mario Coiro[5], Raphaël Cornette[9], Juliana H.L. El Ottra[10], Cyril Epicoco[1], Charles S.P. Foster[11], Florian Jabbour[9], Agathe Haevermans[9], Thomas Haevermans[9], Rebeca Hernández[3], Stefan A. Little[1], Stefan Löfstrand[2], Javier A. Luna[12], Julien Massoni[13], Sophie Nadot[1], Susanne Pamperl[2], Charlotte Prieu[1], Elisabeth Reyes[1], Patrícia dos Santos[14], Kristel M. Schoonderwoerd[15], Susanne Sontag[2], Anaëlle Soulebeau[9], Yannick Staedler[2], Georg F. Tschan[16], Amy Wing-Sze Leung[17] & Jürg Schönenberger[2]

Recent advances in molecular phylogenetics and a series of important palaeobotanical discoveries have revolutionized our understanding of angiosperm diversification. Yet, the origin and early evolution of their most characteristic feature, the flower, remains poorly understood. In particular, the structure of the ancestral flower of all living angiosperms is still uncertain. Here we report model-based reconstructions for ancestral flowers at the deepest nodes in the phylogeny of angiosperms, using the largest data set of floral traits ever assembled. We reconstruct the ancestral angiosperm flower as bisexual and radially symmetric, with more than two whorls of three separate perianth organs each (undifferentiated tepals), more than two whorls of three separate stamens each, and more than five spirally arranged separate carpels. Although uncertainty remains for some of the characters, our reconstruction allows us to propose a new plausible scenario for the early diversification of flowers, leading to new testable hypotheses for future research on angiosperms.

[1] Laboratoire Écologie, Systématique, Évolution, Université Paris-Sud, CNRS UMR 8079, Orsay 91405, France. [2] Department of Botany and Biodiversity Research, University of Vienna, Rennweg 14, Vienna 1030, Austria. [3] Instituto de Biología, Universidad Nacional Autónoma de México, Circuito Exterior, Ciudad Universitaria, Coyoacán, México City 04510, México. [4] Department of Evolution and Ecology, University of California, Davis, California 95616, USA. [5] Department of Systematic and Evolutionary Botany, University of Zurich, Zurich 8008, Switzerland. [6] School of Biological Sciences, Royal Holloway, University of London, Egham, Surrey TW20 0EX, UK. [7] Departamento de Ecología, Pontificia Universidad Católica de Chile, Alameda 340, Santiago, Chile. [8] Systematic Botany and Mycology, Department of Biology, University of Munich LMU, Munich 80638, Germany. [9] Institut de Systématique, Evolution, Biodiversité, Muséum National d'Histoire Naturelle, UMR 7205 ISYEB MNHN/CNRS/UPMC/EPHE, 57 rue Cuvier, CP39, Paris 75005, France. [10] Laboratório de Sistemática Vegetal, Instituto de Biociências, Universidade de São Paulo, Rua do Matão, 277. Cidade Universitária, São Paulo 05508-090, Brazil. [11] School of Life and Environmental Sciences, University of Sydney, Sydney, New South Wales 2006, Australia. [12] Royal Botanic Garden Edinburgh, 20A Inverleith Row, Edinburgh EH3 5LR, UK. [13] Institute of Microbiology, ETH Zurich, Zurich 8093, Switzerland. [14] Centre for Ecology, Evolution and Environmental Changes, Faculdade de Ciências, Universidade de Lisboa, Lisboa 1749-016, Portugal. [15] Organismic and Evolutionary Biology, Harvard University, 26 Oxford Street, Cambridge, Massachusetts 02138, USA. [16] Department of Plant and Environmental Sciences, University of Gothenburg, Carl Skottsbergs gata 22B, Göteborg 413 19, Sweden. [17] School of Biological Sciences, The University of Hong Kong, Pokfulam Road, Hong Kong, China. Correspondence and requests for materials should be addressed to H.S. (email: herve.sauquet@u-psud.fr) or to J.S. (email: juerg.schoenenberger@univie.ac.at).

Flowers are the reproductive structures of angiosperms (flowering plants), which represent ca. 90% of all living land plants and upon which most of terrestrial life depends, either directly or indirectly. However, flowers are a relatively recent evolutionary innovation on the geological timescale of plant diversification. The most recent common ancestor of all living angiosperms likely existed ~140–250 million years ago[1–3]. In contrast, the most recent common ancestor of all extant seed plants (that is, angiosperms and gymnosperms) is estimated to have lived ~310–350 million years ago[4,5]. A key question in evolutionary biology concerns the origin of the angiosperms and of their most important defining structure, the flower[4,6–12]. To address this problem, there are three complementary approaches[7]. The first is to study the fossil record and attempt to identify the closest extinct relatives of angiosperms[4,6]. The second is to seek answers in the growing body of evolutionary developmental genetic (evo-devo) studies on the reproductive structures of living angiosperms and gymnosperms[8,11,13,14]. The third approach, which we apply here using a massive new data set and state-of-the-art analytical methods, is to infer the structure of ancestral flowers using the distribution of floral traits among extant angiosperms, the latest estimates of their phylogeny and models of morphological evolution. This approach allows us to uncover important clues on the origin and subsequent diversification of the flower by providing estimates of what flowers were like at key points in time.

Previous attempts to reconstruct the ancestral flower using a modern phylogenetic framework of angiosperms[15–17] have improved our understanding of ancestral floral traits, such as the ancestral structure of the carpel[18]. However, several essential aspects of the ancestral flower have so far remained unresolved, due to particularly confounding variation in floral structure among the earliest diverging lineages of angiosperms[18–20]. For instance, it was still unknown whether the ancestral flower was unisexual or bisexual[21]. Furthermore, although the reconstruction of the ancestral flower has received some attention, the more general question of its subsequent early evolution and diversification has been little addressed in recent years[9,20,22]. In addition, previous efforts were limited by taxon sampling and the lack of model-based approaches to address these questions.

Here we present the largest data set of floral traits ever assembled (13,444 referenced data points), sampling 792 species from 63 orders (98%) and 372 families (86%) of angiosperms. Using chronograms from molecular dating analyses calibrated with 136 fossil constraints[1], we provide the first model-based reconstructions of ancestral flowers at the deepest nodes in the phylogeny of angiosperms. We infer ancestral states for 27 floral traits using three approaches: maximum parsimony (MP), maximum likelihood (ML) and a reversible-jump Markov Chain Monte Carlo (rjMCMC) Bayesian approach that allows simultaneous exploration of multiple models of morphological evolution. In addition, each analysis was replicated using alternative hypotheses for early angiosperm phylogeny (for example, whether *Amborella* alone or *Amborella* and Nymphaeales together are the sister group of all remaining

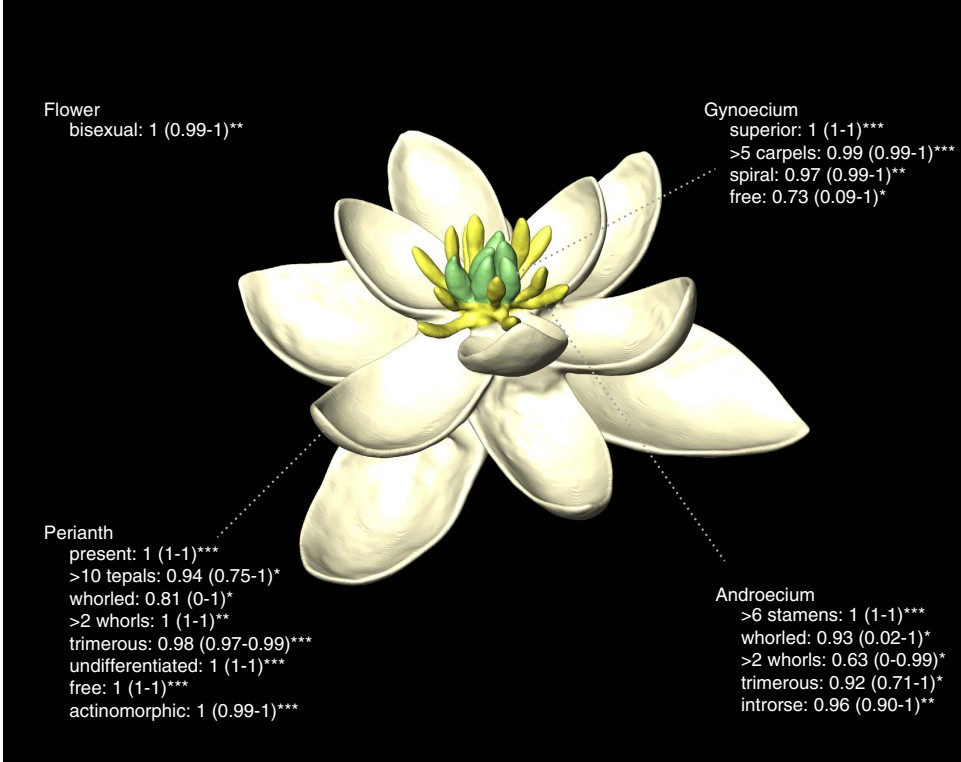

**Figure 1 | Three-dimensional model of the ancestral flower reconstructed from our analyses.** Here we provide the states with highest mean posterior probability and their associated credibility intervals from the reversible-jump Bayesian analysis of the C series of trees, which takes into account all forms of uncertainty (parameters, tree, branch times, model). States marked with three asterisks (***) indicate high confidence and consistency across methods of reconstruction (for example, perianth present, undifferentiated and actinomorphic). Other states need to be interpreted with caution as their reconstruction was either associated with higher uncertainty (for example, perianth phyllotaxis, number of stamen whorls) or inconsistent across methods (for example, sex reconstructed as equivocal with parsimony). The colours, shapes and relative sizes of organs were not inferred from our analyses and were chosen here for artistic reasons. The exact number of organs could not be reconstructed precisely. Minimum numbers were chosen for this representation, but reconstructions with more floral organs are also compatible with our results (for further details, see Supplementary Discussion, section 'Reconstructing the ancestral flower'). A rotating version of this model is provided as Supplementary Movie 1.

angiosperms) and two alternative estimates for the age of the angiosperms, which remain highly debated topics (Supplementary Discussion)[1,2,4,23]. We found that our results are generally robust and unaffected by the choice of ancestral state reconstruction method, alternative phylogenies and different divergence time estimates. However, model-based methods (ML and Bayesian) resolve some long-standing questions where parsimony continues to give equivocal answers.

## Results

**The reconstructed ancestral flower.** We infer that the flower of the most recent common ancestor of all living angiosperms (hereafter referred to as the ancestral flower) was most likely bisexual and had an undifferentiated perianth of more than ten tepals, an androecium of more than ten stamens, and a gynoecium of more than five carpels. We also infer that the perianth and the androecium probably had whorled phyllotaxis with three organs per whorl. Taken together, these numbers imply at least four whorls in each organ category (Fig. 1; see Supplementary Data 1 and Supplementary Discussion for estimates of uncertainty associated with ancestral states). Further, we show that the perianth was radially symmetric (actinomorphic), the stamens had introrse anthers (that is, shedding their pollen towards the centre of the flower), the carpels were superior and most likely spirally arranged, and all floral organs were free from each other. In spite of similarities with some extant flowers, there is no living species that shares this exact combination of characters. This implies that all extant flowers, including those of the earliest-diverging lineages of angiosperms (for example, *Amborella* and Nymphaeales), are derived in several aspects[24]. In particular, the model-based answer to the much-debated question of sex evolution in angiosperms as a whole shows that the ancestral flower was bisexual and confirms that the functionally unisexual flowers of *Amborella* are derived (Fig. 2 and Supplementary Discussion).

We also evaluated the level of correlation among floral traits and its impact on reconstructed ancestral states. We found significant support for correlated evolution in 40–48% of the pairs tested (Table 1), a result consistent with previous studies of floral integration[25–30]. However, accounting for these correlations does not substantially affect the results obtained from analyses of individual traits (Supplementary Data 2 and Supplementary Discussion).

**Uncertainty in ancestral state reconstructions.** Estimating features of the ancestral flower is a difficult task, because there are neither suitable outgroups for direct comparison[4,10] nor fossil flowers known from the time period when this ancestor existed[31]. In this study, we make these inferences based on the distribution of traits in extant angiosperms and their phylogenetic relationships, and, for the first time, methods using explicit models of stochastic evolution for morphological characters. While these analyses help us resolve long-standing ambiguities (for example, whether the ancestral flower was bisexual or unisexual) and reconstruct ancestral flowers at internal key nodes rarely assessed in previous work (for example, Pentapetalae), such reconstructions necessarily come with limitations and some uncertainty. However, it should be possible to quantify this uncertainty.

Through our detailed comparison of three reconstruction methods, five series of trees (each sampling >1,000 chronograms obtained from fossil-calibrated divergence time analyses in BEAST), two timescales for the angiosperms and many models of morphological evolution, we found that reversible-jump Bayesian methods perform best at measuring uncertainty in

ancestral state reconstruction, whereas ML nearly always suggested misleadingly high confidence (Supplementary Discussion). For this reason, 95% credibility intervals (CIs) obtained from the reversible-jump Bayesian analyses are reported throughout this study (Fig. 1 and Supplementary Data 1). This is an important step forward because previous higher-level studies of floral evolution focused almost exclusively on parsimony reconstructions and lacked any assessment of uncertainty associated with ancestral states. Furthermore, early work on ancestral state reconstruction suggested a positive relationship between uncertainty and node depth[32], which would predict that all ancestral states reconstructed for the root of our angiosperm tree should be uncertain. Interestingly, we found that this is not always true (about half of the floral traits examined yielded highly confident estimates; Fig. 3 and Supplementary Discussion), although we observe that focal nodes nested in Monocotyledoneae and Eudicotyledoneae are on average reconstructed with higher confidence than deeper nodes.

**A new scenario for the early evolution of flowers.** Our study provides the first tentative evidence that the ancestral flower of all angiosperms most likely had a perianth (tepals) and an androecium (stamens) organized in whorls, rather than in a spiral. Although reconstruction of ancestral floral phyllotaxis proved relatively uncertain in this study (Supplementary Discussion), as in previous work based on parsimony alone[18–20], the implications of our result are important to consider for two reasons. First, the idea that whorled phyllotaxis of floral organs always evolved from spiral phyllotaxis is still prevalent among botanists. Our analyses provide the most comprehensive evidence so far that the opposite is more likely within crown-group angiosperms (this does not preclude the possibility that the ancestral flower was itself derived from a spiral ancestor further down the stem lineage of the group). Second, this result, if correct, would imply that the early evolution of angiosperm flowers was marked by successive reduction of the number of whorls in both the perianth and the androecium (Fig. 4). The vast majority of angiosperm flowers are characterized by two perianth whorls and one or two stamen whorls (Fig. 5)[22]. Our results suggest two different evolutionary pathways for the reduction in number of whorls in early angiosperm evolution: reduction by loss of entire whorls (Magnoliidae, Monocotyledoneae) or reduction by merging of whorls concomitant with an increase in the number of organs per whorl (Pentapetalae) (Supplementary Discussion). This scenario has implications for comparative evo-devo studies of floral structure across angiosperms, prompting a re-examination of available evidence and interpretations of ABCE model variants[13,33]. In particular, this scenario implies that the two perianth whorls of Monocotyledoneae could be homologous with the corolla (inner perianth whorl) of Pentapetalae (Fig. 4 and Supplementary Discussion), suggesting that the 'sliding boundary' ABCE model of Liliaceae could in fact be a conserved *Arabidopsis* ABCE model expressed in reduced flowers lacking the ancestral two outermost perianth whorls. However, other alternatives exist, including one where the two perianth whorls of Monocotyledoneae are homologous with the calyx (outer perianth whorl) of Pentapetalae by loss of the ancestral two innermost perianth whorls.

What does this scenario of early whorl reduction tell us about the evolutionary forces at play? We propose that early reduction in the number of whorls of ancestral flowers presented selective advantages that eventually led to the extinction of its original, multiparted floral groundplan. First, both the protective function of the perianth and its role in pollinator attraction could be achieved through fewer organ whorls, potentially explaining the progressive loss or merging of whorls. Second, it is possible that a

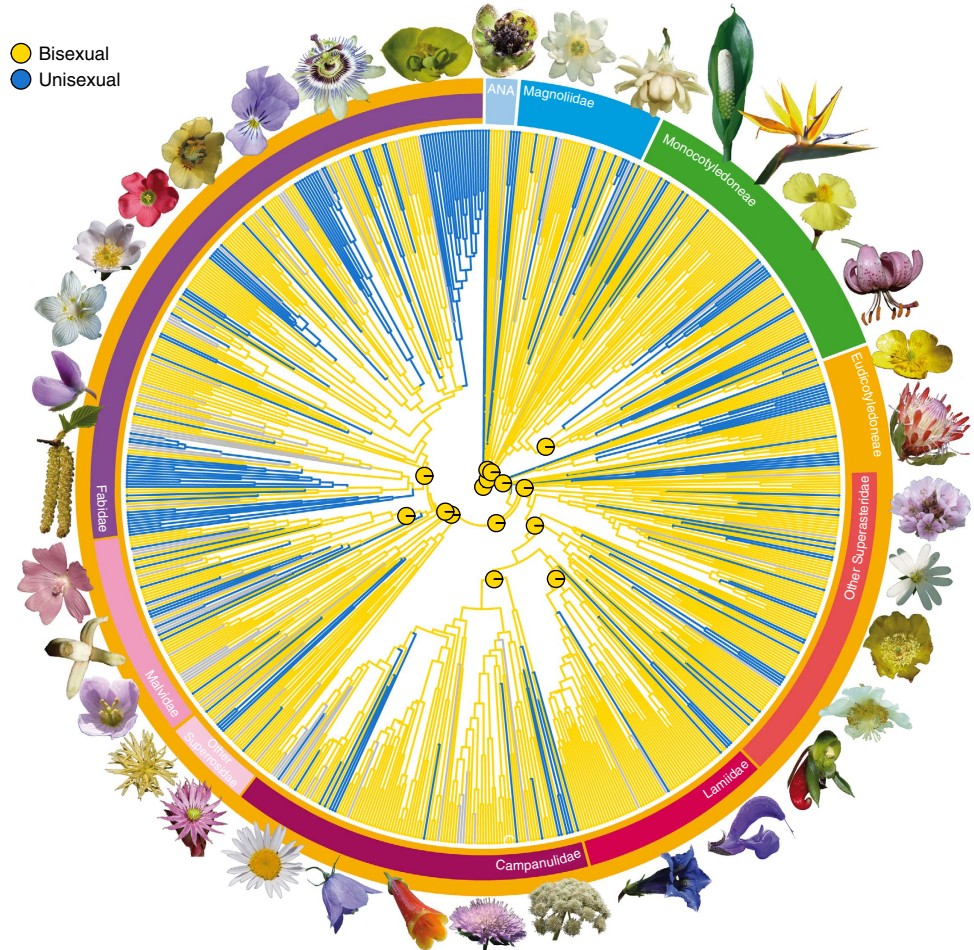

**Figure 2 | Maximum likelihood ancestral state reconstruction of functional sex of flowers in angiosperms.** Our results show that bisexual flowers are ancestral and that unisexual flowers evolved many times independently. The pie charts at the centre of the figure indicate the proportional likelihoods for reconstructed ancestral states at 15 key nodes (here we illustrate character 100_A on the maximum clade credibility tree from the C series; for complete results, see Supplementary Data 1 and Supplementary Data 14–23). The photographs illustrate the diversity of angiosperm flowers (photographs by H.S., Y.S., J.S. and M.v.B.).

reduced number of perianth whorls facilitated the divergence and canalization of genetic programs among whorls, leading to the strong perianth differentiation into sepals and petals that is characteristic of most members of Pentapetalae[13]. Third, a reduced number of whorls may have been a prerequisite for secondary elaboration of floral structure (for example, bilateral symmetry, fusion of organs; Fig. 5), which led to the wide diversity of floral forms and pollination strategies observed in contemporary flowers[25]. In particular, a reduced number of whorls may have been selected for because it facilitated the close spatial and functional association of organs leading to a higher level of functional complexity[34]. This process, known as synorganization, is thought to have increased pollination efficiency and helped trigger some of the most spectacular radiations in angiosperms, such as the Asteraceae and Orchidaceae[35].

From a functional perspective, it may seem difficult to explain why the hypothesized ancestral flower had more perianth organs than most extant flowers. It is plausible that this property is a contingent result of the series of evolutionary transformations (as yet unknown) that led to the ancestral flower from its seed plant ancestors, rather than representing an optimal structure. We suggest that the ancestral flower may in fact have been labile with respect to the number of perianth and androecium whorls and thus the total number of organs in each category. More stable

patterns in the early evolutionary history of angiosperms evolved either by reduction in the number of whorls (as outlined above) or by a transition to spiral phyllotaxis, which has been argued to provide an optimal spatial arrangement in flowers with many organs[36]. Thus, under our scenario, we interpret the entirely spiral flowers of lineages such as *Amborella*, Austrobaileyales and Calycanthaceae as alternative trajectories in floral evolution from a multiparted, whorled ancestor.

## Discussion

In principle, the fossil record could inform us about the plausibility of our reconstructed ancestral flower and our proposed scenario for its subsequent diversification. The oldest confirmed fossil flowers are no older than 130 Ma[6,31,37,38], whereas estimates for the most recent common ancestor of all living angiosperms (that is, the age of our reconstructed ancestral flower) range between 140 and 250 Ma[1–3]. By the time of the first extensive record of fossil flowers in the late Aptian and Albian (100–120 Ma), fossils indicate that the radiation of angiosperms had proceeded well into Nymphaeales, Magnoliidae, Chloranthaceae, early-diverging eudicots and early-diverging monocots[6,31,39], as also implied by our scenario (Fig. 4). Many angiosperms in these Aptian–Albian floras and the few known older ones had simple flowers[6,37–39], which both the present and previous analyses[18,20] interpret as secondarily reduced. However,

**Table 1 | Matrix of all pairwise correlations tested among binary floral traits.**

| | 100_A | 102_B | 201_A | 201_C | 230_A | 231_B | 232_C | 234_A | 204_A | 207_A | 301_C | 330_A | 331_B | 332_C | 305_A | 311_B | 312_A | 313_B | 401_C | 400_A | 403_A | 411_B |
|---|---|---|---|---|---|---|---|---|---|---|---|---|---|---|---|---|---|---|---|---|---|---|
| 100_A. Functional sex of flowers | . | 0.16 | 0.97 | 1 | 0.59 | 0.04 | 0.17 | 1 | 1 | 0.99 | 0.56 | 0 | 0.89 | 0.37 | 0.04 | 0.3 | 0.05 | 0.59 | 0.09 | 0.19 | 0.97 | 1 |
| 102_B. Ovary position | 0 | . | 0.56 | 0.27 | 0.98 | 0.27 | 0.4 | 0.04 | 0.8 | 0.97 | 0.05 | 0.12 | 0.3 | 0.44 | 0.68 | 1 | 0.14 | 0.03 | 0.59 | 0.06 | 1 | 1 |
| 201_A. Perianth presence | Inf | 0 | . | 1 | 1 | 1 | 1 | 1 | 0.99 | 0.92 | 1 | 0.78 | 1 | 1 | 0.02 | 1 | 1 | 0.92 | 0.8 | 0.23 | 0.61 | 0.05 |
| 201_C. Nr of perianth parts | 2.1 | 0.09 | Inf | . | 0 | 1 | 1 | 1 | 1 | 0.43 | 1 | 1 | 0.05 | 1 | 0.74 | 1 | 0.99 | 0.24 | 1 | 1 | 1 | 0.67 |
| 230_A. Perianth phyllotaxis | 0.08 | Inf | 0.01 | Inf | . | 0.92 | 1 | 0.95 | 1 | 0.42 | 1 | 0.99 | 0.01 | 1 | 0.39 | 1 | 0.89 | 1 | 1 | 1 | 1 | 0.59 |
| 231_B. Nr of perianth whorls | 0 | 0.02 | Inf | Inf | 4.19 | . | 0.91 | 1 | 1 | 0.49 | 0.58 | 1 | 0.98 | 0.97 | 1 | 0.7 | 0.89 | 0.99 | 0.95 | 0.94 | 1 | 1 |
| 232_C. Perianth merism | 0 | 0.08 | 0 | Inf | 0 | 8.39 | . | 0.5 | 0.99 | 0.19 | 0.83 | 0.95 | 1 | 1 | 0.13 | 0.91 | 0.85 | 0.82 | 0.97 | 0.6 | 1 | 0.08 |
| 234_A. Perianth differentiation | Inf | 0 | Inf | Inf | Inf | Inf | 0.03 | . | 1 | 1 | 0.72 | 0.76 | 1 | 0.86 | 0.25 | Inf | 0.07 | 0.76 | 0.06 | 0.92 | 1 | 1 |
| 204_A. Fusion of perianth | 0 | 0 | 2.1 | 0.73 | Inf | Inf | 0.04 | Inf | . | 0.89 | 1 | 0.86 | 0.73 | 0.45 | 0.09 | 0.87 | 1 | 0.41 | 0.99 | 0.99 | 0.99 | 1 |
| 207_A. Symmetry of perianth | 1.4 | 0.09 | 0.11 | 0.01 | 0.01 | 0.73 | Inf | 0.03 | 0.05 | . | 0.28 | 0.35 | 0.88 | 0.14 | 0.06 | 0.71 | 0.13 | 1 | 0.51 | 0 | 0.84 | 0 |
| 301_C. Nr of fertile stamens | 0 | 0 | 0 | Inf | 16.77 | 0.1 | 0 | 0 | Inf | Inf | . | 0.92 | 1 | 0.99 | 0.57 | 0.68 | 0.65 | 0.99 | 1 | 0.81 | 1 | 0 |
| 330_A. Androecium structural phyllotaxis | 0.01 | 0.01 | 0 | Inf | Inf | Inf | 8.39 | Inf | 0.07 | 8.39 | | . | 0.99 | 0.98 | 1 | 0.89 | 1 | 0.88 | 1 | 0.99 | 1 | |
| 331_B. Nr of androecium structural whorls | Inf | 0 | 0 | Inf | 8.39 | Inf | 0.05 | 0.08 | 0.26 | 0.03 | Inf | 0.26 | . | 1.01 | 1 | 0.97 | 0.99 | 0.75 | 0.94 | 0.76 | 0.97 | 0.96 |
| 332_C. Androecium structural merism | 0 | 0 | 0 | Inf | 0.76 | 0.67 | Inf | 0.2 | 0.02 | 0.01 | 0.07 | 0.03 | 0 | . | 0.32 | 0.94 | 0.99 | 1 | 0.98 | 1 | 0.23 | 0.05 |
| 305_A. Filament | 0 | 0 | 0.04 | 0.01 | Inf | Inf | 0.01 | 0 | 0.01 | 0.21 | Inf | Inf | 0.03 | Inf | . | 0.1 | 0.41 | 1 | 0.53 | 1 | 1 | 1 |
| 311_B. Anther orientation | 0.01 | Inf | Inf | 0.15 | Inf | Inf | Inf | 0.21 | 0 | 0.01 | Inf | Inf | Inf | Inf | 0 | . | 0.99 | 0.91 | 0.68 | 0.16 | 0.47 | 0.23 |
| 312_A. Anther attachment | 0.01 | 16.77 | 16.77 | Inf | 8.39 | 0.11 | 0 | 0 | 0 | Inf | 16.77 | Inf | 0.43 | 0.14 | 0.42 | | . | 1 | 0.86 | 0.45 | 0.99 | 0.03 |
| 313_B. Anther dehiscence | Inf | 0 | 0 | 0.01 | 5.59 | 0.25 | 0.08 | Inf | 0.03 | 5.59 | 0.06 | 0.23 | 0.07 | 0.12 | Inf | Inf | Inf | . | 0.89 | 1 | 0.99 | 0.86 |
| 401_C. Nr of structural carpels | 0 | 0.01 | 0 | 0.08 | 0.03 | 1.86 | 0.37 | 0.01 | 16.77 | 0.06 | Inf | 0.36 | 0.08 | 1.52 | 1.52 | Inf | Inf | 0.05 | . | 0.85 | 1 | 0.09 |
| 400_A. Gynoecium phyllotaxis | 8.39 | 0.2 | Inf | Inf | Inf | 0.03 | Inf | 16.77 | 16.77 | 0.08 | Inf | 16.77 | Inf | Inf | 0.05 | Inf | Inf | | | . | 1 | 1 |
| 403_A. Fusion of ovaries | Inf | 8.39 | 0 | 0.11 | Inf | Inf | 0.01 | 0.04 | Inf | 0.33 | 1.52 | 16.77 | 1.12 | 0.03 | Inf | Inf | 4.19 | Inf | Inf | Inf | . | 0.45 |
| 411_B. Nr of ovules per functional carpel | Inf | Inf | 0 | 0.22 | 0 | 0 | 0 | 0 | 0 | 0.01 | 0 | 0.01 | 0.02 | 0 | 0 | 0 | 0 | 0 | 0 | 0 | 0.01 | . |

The upper part (above the diagonal) corresponds to maximum likelihood analyses and reports the cumulative Akaike weights of correlated models (supported when ≥0.95). The lower part (below the diagonal) corresponds to the reversible-jump Bayesian analyses and reports the Bayes Factor comparing dependent (correlated) to independent (uncorrelated) models (supported when ≥3; Inf = infinite, indicating no support for uncorrelated evolution). Highlighted cells correspond to strongly supported correlations. For all details and a discussion of these results, see Supplementary Data 2 and Supplementary Discussion. Nr, Number.

the record is consistent with our reconstruction in that late Aptian and Albian flowers with whorled and often trimerous phyllotaxis are more diverse than those with spiral phyllotaxis, and in that no fossils with the typical Pentapetalae pattern of five sepals and five petals are known until the latest Albian[40]. Therefore, although there is a probable time lag in fossil preservation of the earliest angiosperm lineages, the sequence of origin of floral traits in the fossil record is largely consistent with our reconstructed initial stages of floral evolution.

The origin of the angiosperm flower remains among the most difficult and most important unresolved topics in evolutionary biology[4,6–11]. The growing understanding of the distribution of floral traits in both fossil and extant taxa, and the availability of modern analytical tools will be crucial in this long-standing debate. Because our approach cannot reconstruct events that occurred on the stem lineage of angiosperms, our study does not address the origin of the flower directly, but it does provide a novel and detailed picture of the flower of the most recent ancestor of all living angiosperms as well as the earliest steps of the subsequent floral diversification. These results are a major step forward for understanding the origin of floral diversity and evolution in angiosperms as a whole. Progress in reconstructing the evolutionary steps that gave rise to the flower of the most recent common ancestor may require new fossil discoveries, especially along the stem lineage of angiosperms[31], or new breakthroughs in evo-devo research[14] and related emerging fields[41].

## Methods

**Phylogenetic analyses.** Ancestral state reconstruction using model-based methods requires a phylogenetic tree with branch lengths proportional to time (that is, a chronogram) or to the number of inferred molecular substitutions (that is, a phylogram). We preferred the first option because we did not want to assume a strict correlation of molecular and morphological evolutionary rates. The recent relaxed clock molecular dating analysis of Magallón et al.[1] was chosen as the starting point for this study because it was calibrated with the largest number (136) of well-justified fossil age constraints ever used at this scale, while at the same time including a very large number of terminal taxa (792), representing 63 orders (98%) and 372 families (86%) of angiosperms. We also reanalysed this data set in a number of alternative ways to evaluate the impact of various parameters of this dated tree on our analyses.

The A series of analyses refers to the original BEAST analyses of Magallón et al.[1], which provided a maximum clade credibility (MCC) tree, used in our parsimony and ML analyses, and a collection of 1,042 trees sampled from the posterior stationary distribution, which we used for our Bayesian analyses of trait evolution. These trees, however, presented two drawbacks for our analyses. First, their topology had been heavily constrained according to the results of Soltis et al.[16], and thus represented only one of the several alternatives for deep-level relationships in angiosperms. Second, the BEAST analyses had been conducted with a fixed topology, producing a collection of trees that differed in branch lengths (times) but not topology. Thus, integrating phylogenetic uncertainty in our Bayesian analyses of trait evolution was the primary motivation for reanalysing the data set in BEAST without fixing the topology.

The B series of analyses refers to the reanalysis of the data set of Magallón et al.[1] in BEAST 1.8 (ref. 42) without using any topological constraints (that is, topology estimated, not fixed), and with all other parameters equal (see below). These analyses produced trees with Amborella sister to Nymphaeales rather than to all other angiosperms, and with monocots sister to Chloranthaceae + Magnoliidae rather than to Ceratophyllaceae + Eudicotyledoneae (see Supplementary Discussion and Supplementary Fig. 1). Other relationships and divergence times were very similar to those found in the A series, but with some variation among trees of the posterior sample regarding the more weakly supported nodes.

The C series of analyses refers to the same setup as the B series, but with two topological constraints for deep-level angiosperm relationships: (1) Amborella sister to the rest of angiosperms; (2) Monocotyledoneae + Ceratophyllaceae + Eudicotyledoneae together forming a clade (excluding Chloranthaceae and Magnoliidae; Supplementary Fig. 1). These two constraints are supported by the majority of phylogenomic analyses based on complete plastid genomes[17,43–45] and are consistent with the 17-gene analyses of Soltis et al.[16]. The results from the C series were very similar to those of the A and B series (see Supplementary Discussion).

The D and E series were set up with two alternative topological constraints for major clades of angiosperms suggested by recent nuclear phylotranscriptomic analyses (Supplementary Discussion and Supplementary Fig. 1). In the D series, we constrained Chloranthaceae, Magnoliidae, Ceratophyllaceae and Eudicotyledoneae to form a clade[23]. In the E series, we constrained Chloranthaceae and Ceratophyllaceae to be sister taxa[46,47].

In addition, we tested the impact of the age of the angiosperms on our ancestral state reconstructions. The original analyses of Magallón et al.[1] included a narrow age constraint of 136–139.35 Ma on the crown-group age of angiosperms based on a quantitative analysis of the fossil record. The A200, B200, C200, D200 and E200 series refer to the exact same setups as the A, B, C, D and E series, but with this constraint removed, resulting in chronograms with crown angiosperms typically over 200 Ma old.

All new phylogenetic and molecular dating analyses were conducted with BEAST 1.8 (ref. 42) using the same settings, fossil calibrations and protocols as in the A series[1]. For the B series, five independent Markov Chain Monte Carlo (MCMC) runs of different length (up to 20M generations) were conducted, for a total of ca. 71M generations (after discarding the first 2M generations from each run as burn-in). The posterior was resampled every 50K generations to produce a set of 1,412 trees used in the Bayesian trait analyses. For the C series, six runs were conducted for a total of ca. 85M generations, which were resampled every 50K generation to produce a set of 1,706 trees. For each of the D, E, B200, C200, D200 and E200 series of analyses, two runs were conducted for a total of 36M generations, which were resampled every 35K generation to produce sets of 1,028 trees. We note that the effective sample size for some parameters of these analyses

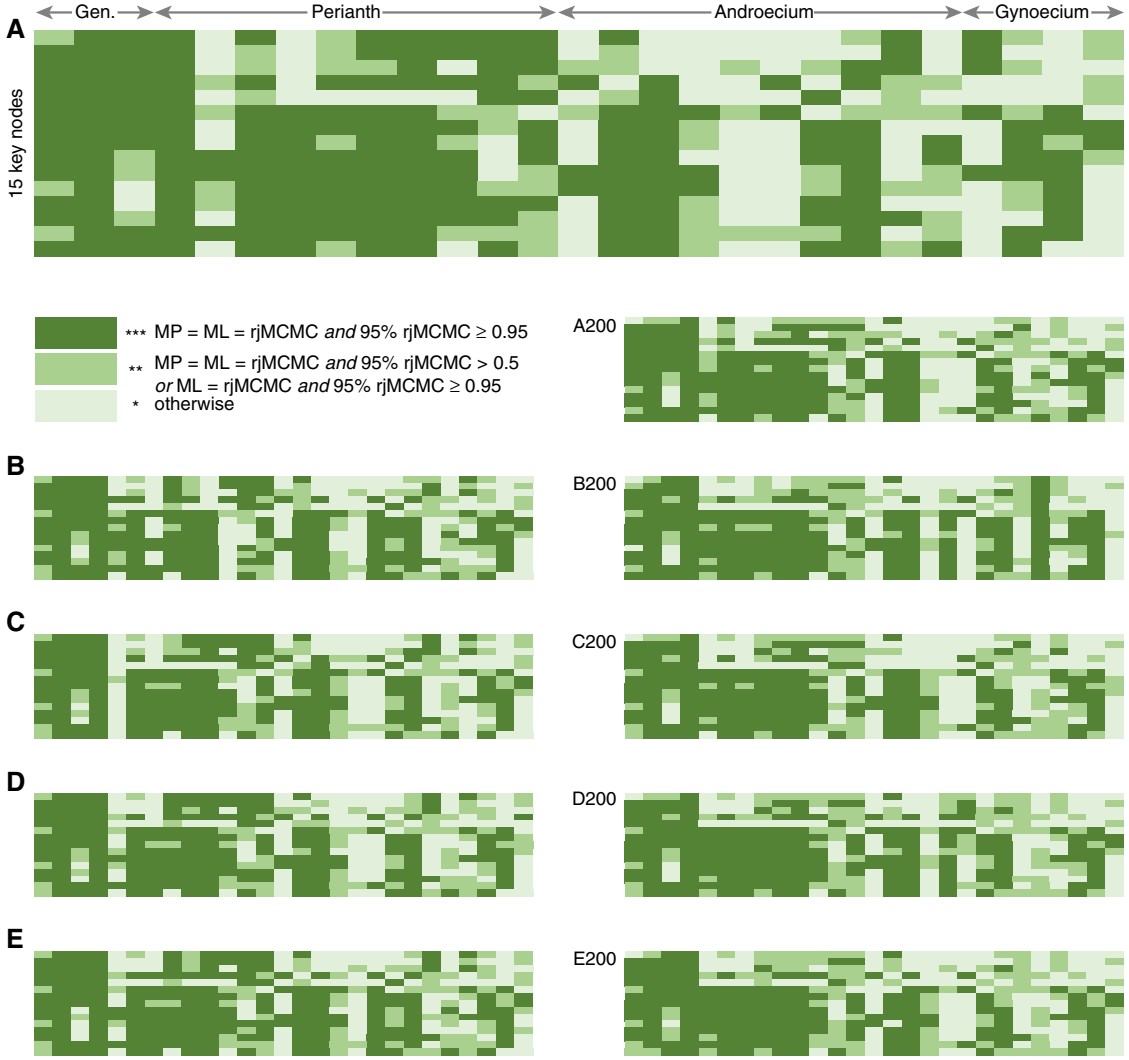

**Figure 3 | Overview of confidence in ancestral state reconstructions across all traits (X axis), focal nodes (Y axis) and sets of trees (panels).** Confidence scores on a three-star scale were attributed for each trait-node combination based on the cross-comparison of MP, ML and rjMCMC results and the lower bound of the rjMCMC credibility intervals (see Supplementary Discussion for details). A more detailed version of this figure with row and column captions is provided in Supplementary Data 1. Light = low confidence; dark = high confidence.

did not all reach 200 as recommended, suggesting that longer runs might be needed for accurate estimation of phylogenetic relationships and divergence times, consistent with the previous finding that this large data set is difficult to analyse with a Bayesian relaxed clock without fixing the topology[1]. We argue that the posterior samples we obtained here are acceptable for the purpose of this study, because the goal of our reanalyses of the Magallón et al.[1] data set was to take into account and evaluate the impact of phylogenetic uncertainty on the results from the A series (the original trees from Magallón et al.[1], with fixed topology). As we report in detail in the Supplementary Discussion, the estimated general topology, divergence times and ancestral states were remarkably similar across tree series (Supplementary Data 1 and Supplementary Tables 1 and 2). However, we recommend caution with the use of these trees for purposes other than this study. The MCC tree from each BEAST analysis is provided as Supplementary Data 3–12.

Terminal taxa in the original molecular data set of Magallón et al.[1] were either species or genera, with different species sampled for different genes. For this study, we transformed the trees of hybrid terminal taxa into trees of species by choosing the species with the most genes sampled for each hybrid (genus-level) terminal taxon. This allowed us to produce trees of 792 species and prepare a matching data set of floral traits for exactly the same species, following a strict exemplar approach (see below). All of our trees also included six outgroup gymnosperm species. Because floral traits are not applicable outside angiosperms (unless controversial homology statements are made), these species were not included in our data set of floral traits and were pruned out of the trees before ancestral state reconstruction. Clade names in this paper follow APG IV[48] and the Angiosperm Phylogeny Website[49] for orders and families, and Cantino et al.[50] and Soltis et al.[16] for all clades above order.

**Data set of floral traits.** We recorded 21 floral traits in 792 species of angiosperms using the collaborative database PROTEUS[51]. The floral traits were chosen and defined to be as broadly applicable as possible. For instance, we do not have a character for the number of petals in this data set, because not all angiosperms have petals and all petals are not necessarily homologous. Instead, we recorded the total number of perianth parts (sepals plus petals, or tepals). All characters are explained and justified in detail in the Supplementary Methods.

Floral traits were recorded from a diversity of published and online sources, including many focused morphological studies and a few personal observations. Each data record in PROTEUS is linked to an explicit source, which allowed us to cross-check, validate or correct many records following initial entry. In total, the data set presented here contains 13,444 floral trait data records obtained from 947 distinct sources. The complete list of records and linked sources (references) is available in Supplementary Data 2.

All primary characters used in data entry were transformed for analysis (discrete characters were simplified and continuous characters were discretized; see Supplementary Methods for justification and details of these transformations). Some characters were transformed in more than one way, leading to a final data matrix of 27 characters and 792 species (Supplementary Data 13). Data files were then exported from PROTEUS in appropriate formats for analysis.

We used a strict exemplar approach for scoring traits, which means that data were only scored for a species if we could confirm that they were observed in this species (that is, we did not use any general family descriptions or make any assumptions that all species of a genus share the same character states). The species were selected because of their inclusion in a recent molecular dating study[1]. Thus, our sample is independent from the floral traits. While this approach is both

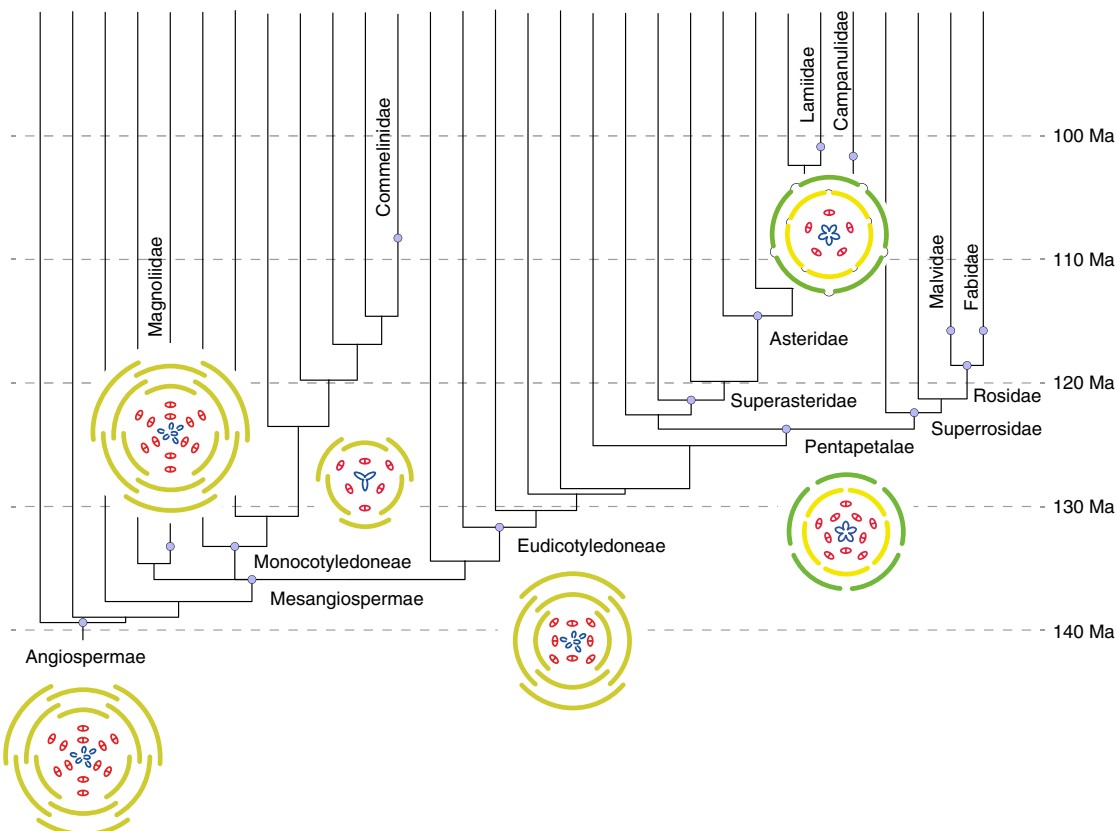

**Figure 4 | Simplified scenario for the earliest phase of floral diversification as inferred from our analyses.** Each floral diagram summarizes the main features of our reconstructed ancestors for key nodes of the tree (for details, see Supplementary Discussion and Supplementary Figs 2–7). This figure only depicts the presumed first 40 million years of floral evolution, without exhaustively representing every new morphology that arose during that time. The absolute timescale provided here corresponds to divergence time estimated with a narrow constraint on the maximum age of angiosperms[1]; relaxing this constraint to reflect alternative studies that yielded older age estimates for angiosperms resulted in nearly identical ancestral reconstructions (see Supplementary Discussion). Note that there is uncertainty associated with some of these reconstructions (especially for Angiospermae, Magnoliidae and Eudicotyledoneae). Therefore, the scenario illustrated here is one of several plausible alternatives and should be taken with caution. Floral diagram colour code: light green = undifferentiated tepals; green = sepals; yellow = petals; red = stamens; blue = carpels.

desirable and suitable for the methods we used, we acknowledge that it implies that our data set does not represent the complete variation of floral traits across all angiosperms. Thus, our study was not designed to reconstruct the finer-scale evolution of flowers near the tips of the tree (for example, within orders), and our results remain conditional on future denser sampling of the angiosperm phylogeny. Our strict exemplar approach also means that data are missing for some traits in some species (total missing data: 27%, including cases of inapplicability). Because missing or inapplicable data are more or less evenly and haphazardly distributed across our tree, and species with such data are in effect pruned out in the ancestral reconstruction analyses, it is unlikely that missing data had a strong impact on our results.

**Ancestral state reconstruction.** Each floral trait was analysed for each series of trees (A, B, C, D, E, A200, B200, C200, D200, E200) using three complementary approaches[52]: MP using the ancestral.pars function of the phangorn 2.0.2 package[53] in R[54], ML using the rayDISC function of the corHMM 1.18 package[55] in R[54], and a Bayesian rjMCMC approach[56,57] using BayesTraits 2 (ref. 58). MP and ML reconstructions were conducted on the MCC tree from each BEAST analysis, whereas Bayesian rjMCMC analyses were conducted on collections of at least 1,000 trees sampled from the posterior stationary distribution from the BEAST analyses. Here, we focus on and report results for 15 key nodes in the phylogeny of angiosperms, corresponding to well-recognized major clades (including Angiospermae, Mesangiospermae, Magnoliidae, Monocotyledoneae, Eudicotyledoneae, Pentapetalae, Rosidae and Asteridae). However, graphical MP and ML reconstructions for the entire tree are available (Supplementary Data 14–23).

For each floral trait, we tested and compared at least two distinct Markov models of discrete character evolution in our ML analyses: the equal rates (ER) or Mk model[59], which assumes a single rate of transition among all possible states, and the all rates different (ARD) or AsymmMk model[60,61], which allows a distinct rate for each possible transition between two states. In addition, we tested two unidirectional models for all binary characters (UNI01 and UNI10: rates from 1 to

0 or 0 to 1, respectively, set to zero)[52,62], a symmetrical model for all multistate characters (SYM: rates equal for transitions between two given states), and three ordered models for all multistate characters derived from quantitative variables (ORD: rates between non-adjacent states set to zero; ORDSYM: symmetrical version; ORDER: single-rate version). Initial tests showed that for some characters, the prior on the root state could affect results in terms of both transition rates and ancestral states[62]. Therefore, we systematically tested both inferences using flat priors[32,63] (equal probability for all states, the default option in most R packages) and a prior with root state frequencies same as equilibrium[64] (we denote such variants with the 'eq' suffix, for example, ARDeq is the implementation of the ARD model with equilibrium root prior), for all models except ER (equilibrium = equal frequencies) and the unidirectional models (root state implied by the model). Although the ARD model might seem more realistic than the more restrictive variants listed above, it may be very difficult to estimate all transition rates accurately, especially for multistate characters. Thus, we tested the fit of these models using the Akaike Information Criterion corrected for sample size, which allowed us to select the model that best fits the data while minimizing the number of parameters[65]. We here report the ML results from the best-fit model.

Bayesian ancestral state reconstruction analyses allowed us to explore three sources of uncertainty not accounted for in ML analyses: transition rate uncertainty, phylogenetic uncertainty and dating uncertainty[57]. In addition, the rjMCMC approach allowed us to explore model uncertainty[56]. This approach is particularly useful where model space is very large, such as for multistate discrete characters (see Supplementary Methods). Each rjMCMC analysis was run in BayesTraits for 10M generations, sampling parameters and ancestral states for 15 key nodes every 100 generations, and starting with an exponential hyperprior with a mean on a uniform interval from 0 to 1. Apparent stationarity was checked in Tracer 1.6 (ref. 66). Discarding the first 1M generations as burn-in was sufficient for all analyses and effective sample size values were nearly always very high (above 200), except for a few particular traits characterized by frequent jumps of the chain between very different models.

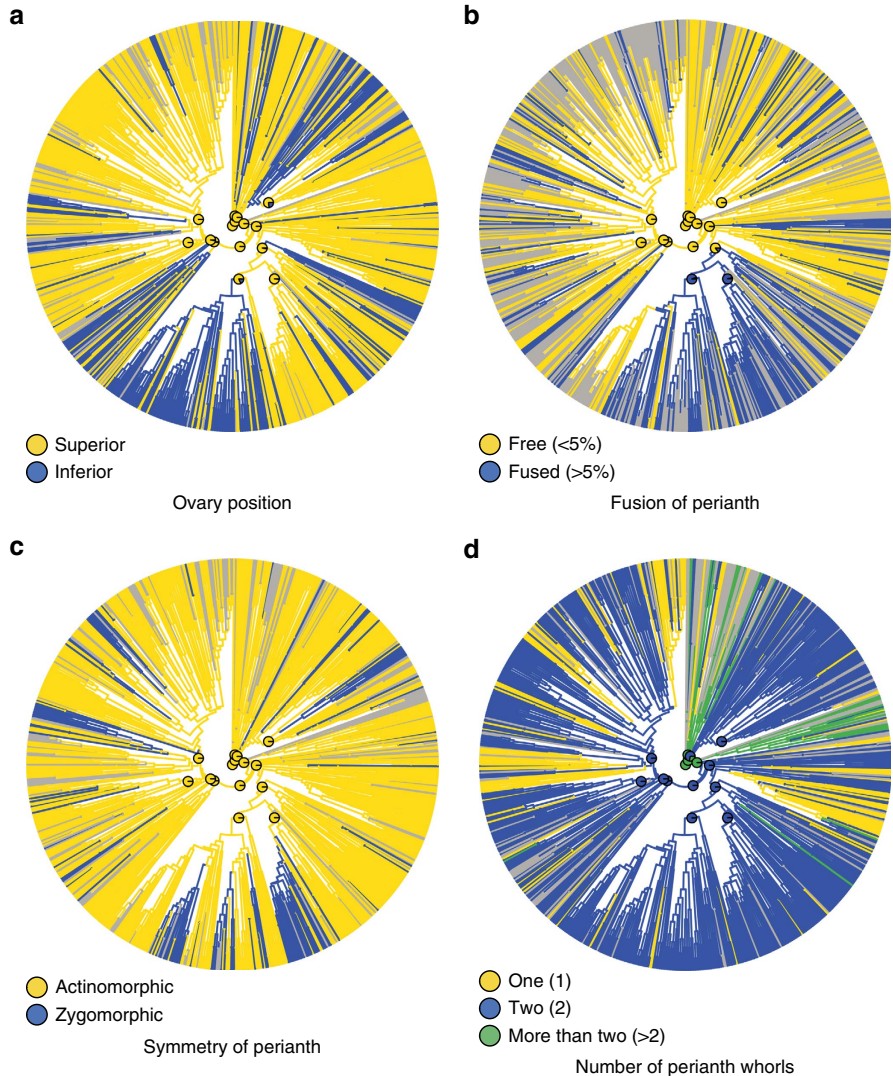

**Figure 5 | Maximum likelihood ancestral state reconstructions of four representative floral characters.** The pie charts at the centre of the figure indicate the proportional likelihoods for reconstructed ancestral states at 15 key nodes (here we optimized the characters on the maximum clade credibility tree from the C series; for complete results, see Supplementary Data 1 and Supplementary Data 14–23). Grey branches denote missing, inapplicable or polymorphic data. (**a**) Ovary position (character 102_B). (**b**) Fusion of perianth (character 204_A). (**c**) Symmetry of perianth (character 207_B). (**d**) Number of perianth whorls (character 231_A).

Here, we report the results from these three analyses at each focal node in the form of the most parsimonious state(s), the most likely state (that is, with highest marginal likelihood), and the state with highest mean probability, respectively (Supplementary Data 1). For the latter (Bayesian rjMCMC), we also report the 95% CI for the probability of the state. In several cases, these CIs are very wide, with probabilities ranging from ca. 0 to 1. Such intervals indicate strong uncertainty in ancestral state reconstructions, where MP and ML can be misleading in showing artificial precision and confidence in the reconstructed ancestral state. For this reason, we refer mostly to the rjMCMC results in this paper and call for caution in interpretation of our results where CIs are very wide. However, for most traits, nodes and trees, the three approaches reconstructed the same ancestral state and rjMCMC CIs were narrow (Supplementary Data 1 and Supplementary Discussion).

**Correlation analyses.** As flowers are highly complex and integrated structures, floral traits are unlikely to evolve independently from one another[25–30]. Although our main goal was not to evaluate the level of morphological integration in flowers, it is possible that such correlations might impact ancestral state reconstructions. However, in contrast to recently developed multivariate approaches for continuous characters[67–69], no comparative method exists yet to account for the potential correlation of more than two discrete characters, unless a drastic simplification of model space is made[25]. In addition, correlated models and analyses have typically been developed for binary characters[56,60]. The stochastic mapping approach to correlation tests allows inclusion of multistate characters, but does not model character correlation and starts at the outset by reconstructing ancestral states

independently at all nodes[70]; it was thus not relevant to our specific objective here. Therefore, we tested correlations among all possible pairs of binary floral traits in our data set. To do so, we first removed redundancies for multiple versions of the same character (Supplementary Methods), and then transformed all multistate characters into binary characters by maintaining the hypothesized ancestral state for the angiosperms as one state and pooling the remaining states as another (for example, for the number of perianth whorls, we analysed one-two whorls versus more than two whorls). We thus obtained a new set of 22 presumably independent characters and analysed all 231 pairwise correlations among these characters (Table 1). Given our observation that reconstructed ancestral states in the single-trait analyses were remarkably consistent across the 10 series of phylogenetic trees (see Supplementary Discussion), we conducted all of our correlation analyses using the C series of trees, which best reflects the current consensus on higher-level angiosperm phylogeny and allows us to take into account phylogenetic uncertainty.

As for our single-trait analyses, we used both an ML and a Bayesian rjMCMC approach to test for correlations and their impact on reconstructed ancestral states, using again the rayDISC function of corHMM 1.18 (ref. 55) in R[54] for ML analyses and BayesTraits 2 (ref. 58) for rjMCMC analyses. The ML approach allowed us to test the fit of a small set of combined Markov models (that is, with $4 \times 4$ $Q$ matrices to model all possible transitions among the four possible combined states, excluding dual transitions), including correlated (dependent) and uncorrelated (independent) models[60]. Specifically, for each character pair, we fitted four correlated models (ARDnodual, ARDnodualeq, differing only in the root state prior: see above; SYMnodual, SYMnodualeq) and three uncorrelated models

(ERnodual, UNCORRnodual, UNCORRnodualeq; UNCORRnodual corresponds to the most general, 4-parameter 'independent' model from ref. 56). Using Akaike Information Criterion corrected for sample size, we selected the best-fit model and compared the ancestral combined states reconstructed with those obtained in our single-trait analyses (Supplementary Data 2). As a measure of support for correlation, we report the cumulative Akaike weight of correlated models (Table 1). The rjMCMC approach allowed us to explore the vast space of the 21,146 possible Markov combined models for the evolution of two binary characters, sampling models according to their posterior probability[56], with settings as above (10M generations, sampling every 100 generations). As for single-trait analyses, the ancestral states reconstructed using this approach integrate over model, parameter, tree and dating uncertainty, as measured by the CIs associated with the probability (proportional likelihood) of each state (Supplementary Data 2). Support for correlation is here measured by the Bayes Factor comparing the dependent models to the independent models, rewritten as the ratio of the posterior to the prior odds of the two models[56]: $BF_{DI} = [P(M_D|D)/P(M_I|D)]/[(21146 - 51)/51]$, where $P(M_D|D)$ and $P(M_I|D)$ are the sampling frequencies of dependent and independent models, respectively.

**Data availability.** Summary (MCC) BEAST trees are provided as Supplementary Data 3–12 and a complete list of morphological data records and references (extracted from PROTEUS) is provided as Supplementary Data 13. Additional trees and data files are available from the authors on request.

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

## Acknowledgements

A large part of the floral data set presented here was assembled during the eFLOWER Summer School held at the University of Vienna, 3–10 July 2013. We thank Ursula Schachner for help in organizing this event; Ralf Buchner for set-up of the eFLOWER server; and Purificación López-García, Susanne Renner and Erik Smets for critical input on an earlier draft of this paper. The Faculty of Life Sciences and the Key Research Area 'Patterns and Processes of Plant Evolution and Ecology' of the University of Vienna, and Agence Nationale de la Recherche grant ANR-12-JVS7-0015-01 (MAGNIPHY) to H.S. provided support for the Summer School and continued development of the eFLOWER project. Additional support was obtained from the Austrian Science Fund (FWF; grant P 25077-B16 to J.S.). Open access funding was provided by the University of Vienna.

## Author contributions

H.S., M.v.B. and J.S. conceived and coordinated the study. H.S., M.v.B., E.B., E.B.M., K.B.-H., L.C., M.C., G.C., M.C., J.H.L.E.O., C.E., F.J., T.H., R.H., S.A.L., S.L., J.A.L., J.M., S.N., S.S., C.P., E.R., P.d.S., K.S., A.S., Y.S., G.F.T., A.W.-S.L. and J.S. compiled the floral data set. S.M. provided phylogenetic trees and the molecular data set. H.S. conducted all analyses. C.S.P.F. helped with continuous traits. R.C., A.H. and S.P. built the 3D model of the reconstructed ancestral flower. S.M., J.A.D. and P.K.E. were important contributors of ideas and data quality management. H.S. and J.S. wrote the first draft of the paper, with subsequent contributions from most coauthors.

## Additional information

**Competing interests:** The authors declare no competing financial interests.

