## [Peer Review File · Nature Communications]

Reviewers' comments:

Reviewer #1 (Remarks to the Author):

Review of "The ancestral flower of angiosperms and its early diversification by H. Sauquet et al.

This paper represents a significant landmark for botanists interested in the origin and evolution of the angiosperm flower. It employs several major analytical innovations, and presents major new findings on the earliest flower. It is carefully done, comprehensive in scope, and has major implications for botanical research on floral diversity and evolution. It is therefore likely to be read by and useful to a broad audience of plant scientists.

The analytical innovations employed here include the use of advanced model-based methods to carefully reconstruct ancestral floral states, using a large and well-populated data set that is constructed and presented here, that mostly focuses on species-level observations. The data comprise a novel and very large compendium of floral information that has not been pulled together before.

Previous efforts to reconstruct the ancestral flower used parsimony for inferring ancestral states. Although this method often works well, it has a limited capacity to deal with tree uncertainty, no capacity to deal with branch-length information (e.g., using fossil-calibrated chronograms, as here), and no real capacity to deal with model uncertainty. Regarding the latter, although parsimony 'model' variants are available, there is no objective way to decide on which to use. In contrast the model-based methods used here allow this to be done in an objective manner – I particularly like the diversity of methods and approaches explored (including the use of reversible jump MCMC approach accounting for model uncertainty and major aspects of tree uncertainty). Significantly and reassuringly, the results presented here are largely insensitive to the diverse reconstruction approaches used, although, as expected, parsimony (the weakest approach for a question like this) provides the most equivocal results.

Given that we now have reasonably accurate angiosperm-wide trees (largely consistent among multiple lines of evidence) to reconstruct character evolution on, and increasingly powerful model-based methods, the time is really ripe for this kind of analysis. Previous efforts scored character states for higher-level clades that relied somewhat subjectively on researcher expertise; decisions on root-node states for individual clades made in those studies, even if locally accurate, were also not necessarily globally optimal. I think it is better to include a representative sampling of individual species -- as done here -- with character scorings done per species, and to let the analyses grapple with variation at that level.

The study is a landmark research advance in the field, that provides a search image for what crown angiosperm flowers must have looked like to paleobotanists considering the fossil record (and notably, the predictions made here are basically consistent with the fossil record), in addition to a series of testable predictions for evo-devo researchers (e.g., on

contrasting paths of floral reduction in monocots and magnoliids vs. pentapetaloids).

Minor concerns

(1) As the authors know, the current analysis reconstructs the first flower of crown angiosperms. Earlier flowers no doubt existed on the long angiosperm stem (perhaps near-crown taxa that would be readily identified as angiosperms if alive today based on their arrays of apomorphies). These earlier flowers are inaccessible using phylogenetic analyses with living taxa alone. I think it would be very useful to indicate some thoughts on how we might do this larger task, for the general reader, perhaps adding a clause like this on l. 141 "... the flower directly, which can only come from further integration of fossil, evo-devo and phylogenetic evidence, ...".

(2) l. 76. Which living species are closest in terms of being 'fewest states away' from the ancestral flower? I realize the authors may be reluctant to declare this for fear of misinterpretation (that these angiosperms are "most primitive", an unfortunate leap in logic that some readers will undoubtedly tend to make). But I think it would be useful to know if handled delicately. Are any of these ANITA-grade taxa?

(3) I'd be careful about calling *Amborella* 'androdioecious', as this term has functional connotations for a pollination biologist (also, 'dioecy' is really a property of populations or species, not individual plants and their flowers). Can a different word or phrase be used (i.e., that it has structurally bisexual flowers that are functionally unisexual)?

(4) l. 105-106. Why did the earliest flower have more protective organs if fewer could work equally well? I realize things have to start somewhere, but why did they start like this? It may be impossible to answer or even speculate about this given available evidence, but it may still be worth posing the question.

(5) I think it is fine to include analyses with ESS values under 200 so long as clearly stated. This is, after all, just a rule of thumb. Perhaps this could be justified a bit better on l. 215 (beyond the caution). One justification may be post-hoc -- that the analysis results are consistent, despite not all crossing this value in some analyses.

(6) Probably say "haphazardly" rather than "randomly" unless you do a test for randomness (I don't think one is needed).

(7) l. 281 onwards. Which of these models, if any, represent the Mk or Mkv models of Paul Lewis (2001 *Syst Biol* 50:913), and the Assym model used in *Mesquite* (which I think comes from Pagel)? If comparable, it would be useful to briefly note this for readers who are more familiar with these well-established methods and packages.

(8) l. 303. It struck me that best-fit models obtained here could be used to objectively justify corresponding parsimony models for individual characters ('Dollo' to indicate basically

irreversible characters; 'Wagner' for ordered characters). Might this lead to less equivocality in the corresponding parsimony results?

(9) I think it is worth emphasizing a bit more in the MS that while this study represents the most comprehensive species-level analysis of this question attempted so far, it is also expected to be somewhat conditional on future improved species sampling. A small example in Nymphaeales (a key clade) is that *Trithuria* is represented here by only one species (*T. submersa*; monoecious) but we know from a well resolved phylogenetic study of the genus that it likely experienced multiple shifts (to unisexuality? not clear) (Iles et al. *AJB* 99:663+). Perhaps cite this study as an example of where future improvements to taxon sampling might be revealing?

Typos, etc

l. 130. "interpreted"

l. 227. Cite APG 4?

l. 312. Delete "reaching of". I also think it's worth saying that it's "apparent" stationarity as you simply can't run these things indefinitely to see if a much longer run would actually be different.

Generally – use 'ca.' (circa) for approximate dates, and '~' for everything else (reverse of what is used in the MS).

Reviewer #2 (Remarks to the Author):

The manuscript by Sauquet et al presents a large scale analysis of the evolution of the angiosperm flower. It is based on high quality data and the authors took particular care to select and characterize key morphological traits defining the flowers. The manuscript is well written and I do not have specific comments on the structure and logical flow of the text. The major criticism that I have on the text itself is that the authors should probably try to make their manuscript more accessible to non botanists. For instance, the end of the abstract has already quite specialized terms in the lines 11-15 ("We reconstruct the ancestral angiosperm flower as bisexual and actinomorphic, with an undifferentiated perianth of four trimerous whorls of free tepals, an androecium of four trimerous whorls of introrse stamens, and a gynoecium of at least six free, superior carpels.") The wording is of course perfectly correct, but could put aside a more generalist reader.

The approach used by the authors was to model using ancestral trait reconstructions the ancestral flowers of the angiosperms. They took particular care to use several approaches to corroborate their results. This is commendable and give a stronger confidence in the results obtained. They also looked at different tree topologies, which is again necessary and well done. I have however two points that are not very well covered in my opinion in the manuscript.

First, the authors provide very little assessment of the error in the ancestral state reconstruction. It is well known that the deeper you go in the phylogenetic tree, the higher

is the uncertainty in the ancestral state estimation. This is true for both maximum likelihood and bayesian approaches. However, the results are not showing this at all (e.g. Fig 1 or 3, for instance). The extended data figure 1, that is suppose to represent this is quite unclear in my opinion and the full text is written as if the basal flowers were estimated without any uncertainty. This should be modified absolutely to better inform the reader about the reliability of your results.

Second, the ancestral state reconstructions are done for every morphological characters independently. This can be justified from a methodological point of view as it is easier to handle the reconstruction for each character one by one, but it makes little sense biologically as the evolution of the different parts of the flower are certainly linked and evolve under some correlation. I fully admit that it is not easy to account for such a dependency between traits, but there are ways to at least assess the level of correlation and discuss it in the manuscript. The authors should look at the papers by Adams and Felice (2014) or Bartoszek et al (2012) for a discussion and some possible ideas on how to do this.

Reviewer #3 (Remarks to the Author):

The authors attempt to reconstruct the ancestral flower using the distribution of floral traits among extant angiosperms. They assemble a dataset of traits from the literature and apply model-based methods to reconstruct ancestral states on alternative phylogenetic trees for angiosperms. They infer that the ancestral flower was most likely bisexual and actinomorphic, with an undifferentiated perianth of four trimerous whorls of free tepals, an androecium of four trimerous whorls of introrse stamens, and a gynoecium of at least six free, superior carpels. This exact combination of characters does not exist in extant angiosperms, and the authors offer scenarios of early floral evolution to explain the floral structure of derived angiosperm clades.

The authors acknowledge that the task of reconstructing the ancestral flower is a difficult one, given the lack of suitable fossils and outgroups, and the variability of floral structure in the basal grade (ANA grade) of angiosperms. Therefore, it is anticipated that they would provide compelling evidence should they arrive at robust conclusions where previous studies have been ambiguous. Disappointingly, the Main Text and supplementary information do not meet these expectations, and are often in conflict in with each other.

Specific examples of conflict include:

1. Ancestral perianth phyllotaxis:

Main text: "Our study provides the first tentative evidence that the ancestral flower of all angiosperms had a perianth (tepals) and an androecium (stamens) organized in whorls, rather than in a spiral."

Supplemental Discussion: "Our analyses also suggest that perianth phyllotaxis (char.

230_A) of the ancestral flower was whorled, although we acknowledge that support for this result remained low.”

The low support should be mentioned in the main text. Moreover, whether their choice to “... score perianth phyllotaxis only at anthesis because developmental data are lacking for most species in our data set.”, affected their character reconstruction should also be explored.

2. Ancestral number of perianth organs:

Main text, line 69 page 4: “We infer that the ancestral flower was most likely bisexual and had an undifferentiated perianth of four whorls of three tepals each...”

Supplemental Discussion, page 26: “... we have decided to tentatively portray the reconstructed ancestral perianth with four whorls of three parts each (12 in total; Fig. 1, Fig. S2), but we acknowledge that other alternatives with fewer or more parts cannot be ruled out based on our analyses.”

Some mention of the uncertainty surrounding the number of perianth organs seems warranted in the main text. Also, and importantly, it is very unclear how the authors arrived at four for the number of whorls of tepals. I cannot find a justification for this number in the main text, and the supplemental discussion seems to suggest that more than two whorls was their most precise reconstruction:

Pg 25: “...we consider it more likely that the ancestral flower had a perianth formed of more than two whorls.”

Pg 26: “we cannot reconstruct the ancestral number of perianth whorls more precisely.”

Possibly, the number four was reached because the total number of perianth parts was reconstructed as “more than 10” in ML and rjMCMC analyses (Supplemental Discussion, page 25) and therefore the fewest trimerous whorls required to accommodate more than 10 tepals is 4. If so, this conclusion seems somewhat arbitrary to me.

3. Ancestral number of stamens:

Main text, line 69 page 4: “We infer that the ancestral flower was most likely bisexual and had ..., an androecium of four whorls of three stamens each...”

Supplemental Discussion, page 26: we tentatively chose to depict the ancestral androecium as formed of four whorls of three parts each (as the perianth), but we acknowledge that other alternatives are possible.

Same comments as for perianth numbers above.

4. Ancestral number of carpels:

Main text, line 69 page 4: “We infer that the ancestral flower was most likely bisexual and had ..., and a gynoecium of at least six carpels...”

Supplemental Discussion, page 27: “The ancestral flower most likely had a superior ovary ... with more than five carpels...”

Similar to the numbers for perianth organs and stamens, it is not entirely clear how the authors reconstruct exactly six carpels in the main text while the modeled character state was “more than five carpels”.

Thus, for all four of these features of the ancestral flower, the manuscript is confusing or unconvincing. Other ancestral features discussed are not controversial and generally support the analyses of earlier publications (e.g, Endress and Doyle 2015).

Reviewers' comments:

Reviewer #1 (Remarks to the Author):

Review of "The ancestral flower of angiosperms and its early diversification by H. Sauquet et al.

This paper represents a significant landmark for botanists interested in the origin and evolution of the angiosperm flower. It employs several major analytical innovations, and presents major new findings on the earliest flower. It is carefully done, comprehensive in scope, and has major implications for botanical research on floral diversity and evolution. It is therefore likely to be read by and useful to a broad audience of plant scientists.

RESPONSE: Thank you very much for your very thoughtful review of our work. We really appreciate all of your comments in support of this study and have answered all of the minor points raised below.

The analytical innovations employed here include the use of advanced model-based methods to carefully reconstruct ancestral floral states, using a large and well-populated data set that is constructed and presented here, that mostly focuses on species-level observations. The data comprise a novel and very large compendium of floral information that has not been pulled together before.

Previous efforts to reconstruct the ancestral flower used parsimony for inferring ancestral states. Although this method often works well, it has a limited capacity to deal with tree uncertainty, no capacity to deal with branch-length information (e.g., using fossil-calibrated chronograms, as here), and no real capacity to deal with model uncertainty. Regarding the latter, although parsimony 'model' variants are available, there is no objective way to decide on which to use. In contrast the model-based methods used here allow this to be done in an objective manner – I particularly like the diversity of methods and approaches explored (including the use of reversible jump MCMC approach accounting for model uncertainty and major aspects of tree uncertainty). Significantly and reassuringly, the results presented here are largely insensitive to the diverse reconstruction approaches used, although, as expected, parsimony (the weakest approach for a question like this) provides the most equivocal results.

Given that we now have reasonably accurate angiosperm-wide trees (largely consistent among multiple lines of evidence) to reconstruct character evolution on, and increasingly powerful model-based methods, the time is really ripe for this kind of analysis. Previous efforts scored character states for higher-level clades that relied somewhat subjectively on researcher expertise; decisions on root-node states for individual clades made in those studies, even if locally accurate, were also not necessarily globally optimal. I think it is better to include a representative sampling of individual species -- as done here -- with character scorings done per species, and to let the analyses grapple with variation at that level.

The study is a landmark research advance in the field, that provides a search image for what crown angiosperm flowers must have looked like to paleobotanists considering the fossil record (and notably, the predictions made here are basically consistent with the fossil record), in addition to a series of testable predictions for evo-devo researchers (e.g., on contrasting paths of floral reduction in monocots and magnoliids vs. pentapetaloids).

Minor concerns

(1) As the authors know, the current analysis reconstructs the first flower of crown angiosperms. Earlier flowers no doubt existed on the long angiosperm stem (perhaps near-crown taxa that would be readily identified as angiosperms if alive today based on their arrays of apomorphies). These earlier flowers are inaccessible using phylogenetic analyses with living taxa alone. I think it would be very useful to indicate some thoughts on how we might do this larger task, for the general reader, perhaps adding a clause like this on l. 141 "... the flower directly, which can only come from further integration of fossil, evo-devo and phylogenetic evidence, ...".

RESPONSE: Thank you for this suggestion. We certainly agree that future discoveries of unequivocal stem-lineage fossil relatives of angiosperms, especially flowers (currently lacking), would definitely represent breakthroughs for our understanding of ancestral flowers. In a recent review published in *Nature Plants*, Herendeen et al. (2017) indeed outlined the significance of such potential future discoveries. We have now added a sentence to the corresponding paragraph to make this clear, citing this review:

"Progress in reconstructing the evolutionary steps that eventually gave rise to the flower of the most recent common ancestor may require new fossil discoveries, especially along the stem lineage of angiosperms³¹, or new breakthroughs in evo-devo research¹⁷ and related emerging fields⁴¹." (Discussion, lines 203-207)

(2) l. 76. Which living species are closest in terms of being 'fewest states away' from the ancestral flower? I realize the authors may be reluctant to declare this for fear of misinterpretation (that these angiosperms are "most primitive", an unfortunate leap in logic that some readers will undoubtedly tend to make). But I think it would be useful to know if handled delicately. Are any of these ANITA-grade taxa?

RESPONSE: This is indeed a question that we asked ourselves, and which is likely to emerge among readers. However, as suggested by the reviewer, there is a great risk of misinterpretation outside the community of phylogenetically minded readers. Indeed, the idea that some living taxa are more primitive than others is still widespread among the general public and some biological disciplines, even though evolutionary biologists have agreed that it is incorrect for several decades. For instance, many biologists, textbooks, class materials, online videos, and mainstream documentaries still present the flower of *Magnolia* as the archetypal flower of angiosperms, an idea inherited from prephylogenetic views on plant evolution dating back to 1907 and strongly propagated by influential authors in the 1960s to 1980s such as Arthur Cronquist. No matter the number of traits that the ancestral flower might share with flowers of extant species of *Magnolia* (several indeed, as it does with other living taxa), we simply cannot portray any extant species as a close match to

an ancestor that lived at least 140 million years ago, as the Reviewer is well aware of. We have discussed this question among co-authors and decided that the risk would be too high among the broad audience we target if we pointed to the living taxa with highest global similarity to our reconstructed ancestral flower, even with appropriate cautionary words.

(3) I'd be careful about calling *Amborella* 'androdioecious', as this term has functional connotations for a pollination biologist (also, 'dioecy' is really a property of populations or species, not individual plants and their flowers). Can a different word or phrase be used (i.e., that it has structurally bisexual flowers that are functionally unisexual)?

RESPONSE: Thank you for pointing this out. We agree and indeed made the difference between functional and structural sex in our dataset and analyses, where we used and compared the two characters (see Supplementary Methods). *Amborella* was scored as "unisexual dioecious (functionally), but structurally androdioecious" as a species, and we agree it was incorrect to refer to "androdioecious flowers" in the main text, which we have corrected (now referring to "functionally unisexual flowers"). In addition, we now realize that the term androdioecious is not commonly used in a structural sense and have corrected the Supplementary Methods and Discussion to this effect. As a result, "androdioecious" is no longer used to describe the structural sex of *Amborella* anywhere in our revised main and supplementary text.

(4) l. 105-106. Why did the earliest flower have more protective organs if fewer could work equally well? I realize things have to start somewhere, but why did they start like this? It may be impossible to answer or even speculate about this given available evidence, but it may still be worth posing the question.

RESPONSE: This is a very important, but difficult question to answer. First, we do not expect nor envision the ancestral flower to have been a "perfect" structure under the environmental conditions of that time, but rather the suboptimal starting point of many future modifications. We have now clarified this point in the first sentence of a new paragraph of the Results (where we elaborate further on the origin of spiral flowers):

"From a functional perspective, it may seem difficult to explain why the hypothesized ancestral flower had more perianth organs than most extant flowers. It is plausible that this property is just the contingent result of the series of evolutionary transformations (as yet unknown) that led to the ancestral flower from its seed plant ancestors, rather than representing an optimal structure." (Results, lines 162-166)

The fact that many organs appear superfluous when fewer might work equally well rather explains why, today, most flowers are characterized by fewer organs (though many exceptions exist). Why the ancestral flower started with many organs can only be answered with future paleobotanical and evo-devo evidence on the origin of the flower, a question our macroevolutionary study cannot answer (as we have now clarified in the Discussion, see above).

(5) I think it is fine to include analyses with ESS values under 200 so long as clearly stated. This is, after all, just a rule of thumb. Perhaps this could be justified a bit better on l. 215 (beyond the caution). One justification may be post-hoc -- that the analysis results are consistent, despite not all crossing this value in some analyses.

RESPONSE: We agree and have added the justification:

“We argue that the posterior samples we obtained here are acceptable for the purpose of this study, because the goal of our re-analyses of the Magallón et al.⁹ data set was to take into account and evaluate the impact of phylogenetic uncertainty on the results from the A series (the original trees from Magallón et al.⁹, with fixed topology). As we report in detail in the Supplementary Discussion, the estimated general topology, divergence times, and ancestral states were remarkably similar across tree series.” (Methods, lines 278-284)

(6) Probably say “haphazardly” rather than “randomly” unless you do a test for randomness (I don’t think one is needed).

RESPONSE: Corrected.

(7) l. 281 onwards. Which of these models, if any, represent the Mk or Mkv models of Paul Lewis (2001 Syst Biol 50:913), and the Assym model used in Mesquite (which I think comes from Pagel)? If comparable, it would be useful to briefly note this for readers who are more familiar with these well-established methods and packages.

RESPONSE: We agree this would be very useful and have added this information (along with relevant references) to the corresponding sentence:

“For each floral trait, we tested and compared at least two distinct Markov models of discrete character evolution in our ML analyses: the equal rates (ER) or Mk model⁵⁹, which assumes a single rate of transition among all possible states, and the all rates different (ARD) or AsymmMk model^{60,61}, which allows a distinct rate for each possible transition between two states.” (Methods, lines 352-356)

(8) l. 303. It struck me that best-fit models obtained here could be used to objectively justify corresponding parsimony models for individual characters (‘Dollo’ to indicate basically irreversible characters; ‘Wagner’ for ordered characters). Might this lead to less equivocality in the corresponding parsimony results?

RESPONSE: We agree this would be very interesting to try. However, the reality is that unidirectional and ordered characters were very rarely selected as best-fit in our analyses and when they were, support was weak. For instance, in the C series, unidirectional models were selected in two out of 15 binary characters (perianth presence and androecium phyllotaxis) and received Akaike weights of 0.35 in both cases (indicating low support).

Ordered models were selected in two out of 10 multistate characters derived from quantitative variables (number of androecium whorls and androecium merism), again with low support (Akaike weights of 0.35 and 0.89). Therefore, we consider that, at least in our study, parsimony analyses informed by these models would be difficult to justify.

(9) I think it is worth emphasizing a bit more in the MS that while this study represents the most comprehensive species-level analysis of this question attempted so far, it is also expected to be somewhat conditional on future improved species sampling. A small example in Nymphaeales (a key clade) is that *Trithuria* is represented here by only one species (*T. submersa*; monoecious) but we know from a well resolved phylogenetic study of the genus that it likely experienced multiple shifts (to unisexuality? not clear) (Iles et al. AJB 99:663+). Perhaps cite this study as an example of where future improvements to taxon sampling might be revealing?

RESPONSE: We certainly agree that our results will require further testing with more densely sampled phylogenies and datasets, even though we think that ancestral states for the fifteen deep nodes presented in this study are unlikely to be affected. We have added a statement to this effect in a sentence of the Methods:

“Thus, our study was not designed to reconstruct the finer-scale evolution of flowers near the tips of the tree (e.g., within orders), and our results remain conditional on future denser sampling of the angiosperm phylogeny.” (Methods, lines 329-331)

The question of *Trithuria* is complex because of the debated interpretation of bisexual reproductive units occurring in some species and the diversity of sexual systems in this small genus. Most of us personally favor the interpretation of these units as inflorescences of unisexual flowers. In this case, all species of *Trithuria* would be unisexual and sampling more species would not change the results (note that in this study we did not attempt to reconstruct ancestral breeding systems, such as bisexual vs. monoecious vs. dioecious species; instead, we focused on the sex of flowers). However, even under the interpretation of bisexual units of *Trithuria* as ‘inverted’ flowers, we expect little effect on our results, given the findings of two recent studies. We have discussed this question in more detail in a new paragraph of the Supplementary Discussion (section “Sex and perianth presence”). Note that we chose not to express our preference for the unisexual flower interpretation because this controversial issue is not essential for this study.

*“While we acknowledge that all our results will require further testing with more densely sampled floral trait datasets and matching dated phylogenies, this result in particular is unlikely to be challenged by increased sampling. Given the position of *Trithuria* (Hydatellaceae) as sister group of all remaining Nymphaeales (Saarela et al. 2007) and the diversity of sexual systems among the 12 species of the genus (Sokoloff et al. 2008), it would be tempting to think that denser sampling of *Trithuria* in particular might have an influence on the results obtained here. Because of the controversial interpretation of cosexual plants with bisexual reproductive units formed of stamens at the centre surrounded by carpels (Rudall et al. 2007), we chose to not score (i.e., leave as missing data) any character that depended on interpretation of these units as flowers or inflorescences (see above under Special cases). Iles et al. (2012) and Anger et al. (2017) recently investigated the ancestral sexual system of *Trithuria* and angiosperms, respectively, using small datasets and phylogenies that included all species of the genus. While the two studies differed in character scoring (two binary vs. one multistate character), both found the ancestral state of*

Trithuria to be ambiguous because of the phylogenetic distribution of the variation in the genus. Therefore, we do not expect that our results would change had we sampled all species of Trithuria and scored them according to either interpretation of the bisexual reproductive units observed in some of the species.” (Supplementary Discussion, page 21)

Typos, etc

I. 130. “interpreted’

RESPONSE: Corrected (with present tense rather than past because the verb also applies to current analyses).

I. 227. Cite APG 4?

RESPONSE: We have now updated the number of orders and families sampled (as well as the orders and families listed in Supplementary Table 1) according to APG IV (2016):

“Here we present the largest dataset of floral traits ever assembled (13,444 referenced data points), sampling 792 species from 63 orders (98%) and 368 (85%) families of angiosperms.” (Introduction, lines 47-49; also updated in Methods, lines 217-218)

Because this study used a strictly exemplar approach, it is immune to taxonomic changes above the species level. The change from APG III to APG IV therefore had no impact on our analyses and results.

I. 312. Delete “reaching of”. I also think it’s worth saying that it’s “apparent” stationarity as you simply can’t run these things indefinitely to see if a much longer run would actually be different.

RESPONSE: Corrected.

Generally – use ‘ca.’ (circa) for approximate dates, and ‘~’ for everything else (reverse of what is used in the MS).

RESPONSE: We are not sure about this distinction and would prefer the copy-editors of *Nature Communications* to decide on this rule according to the journal style, should our manuscript be accepted.

Reviewer #2 (Remarks to the Author):

The manuscript by Sauquet et al presents a large scale analysis of the evolution of the angiosperm flower. It is based on high quality data and the authors took particular care to select and characterize key morphological traits defining the flowers. The manuscript is well written and I do not have specific comments on the structure and logical flow of the text. The major criticism that I have on the text itself is that the authors should probably try to make their manuscript more accessible to non botanists. For instance, the end of the abstract has already quite specialized terms in the lines 11-15 ("We reconstruct the ancestral angiosperm flower as bisexual and actinomorphic, with an undifferentiated perianth of four trimerous whorls of free tepals, an androecium of four trimerous whorls of introrse stamens, and a gynoecium of at least six free, superior carpels.") The wording is of course perfectly correct, but could put aside a more generalist reader.

RESPONSE: Thank you very much for your thorough review and comments, which we found very helpful.

Botanical vocabulary is indeed quite diverse and may appear cryptic to some readers outside the field. This is one of the reasons why we made a special effort to carefully define and explain all of the characters used in our study, in the Supplementary Methods attached to this paper. With respect to the main text, we think that replacing clear, standard botanical terms with more accessible definitions might be misleading to colleagues in the field (because of the risk of varying interpretations) and come at the cost of concision. However, we agree that the Abstract should be accessible to the broadest possible audience and have thus reworded the problematic sentence:

"We reconstruct the ancestral angiosperm flower as bisexual and radially symmetric, with more than two whorls of three separate perianth organs each (undifferentiated tepals), more than two whorls of three separate stamens each, and more than five spirally arranged separate carpels." (Abstract, lines 8-12)

The approach used by the authors was to model using ancestral trait reconstructions the ancestral flowers of the angiosperms. They took particular care to use several approaches to corroborate their results. This is commendable and give a stronger confidence in the results obtained. They also looked at different tree topologies, which is again necessary and well done. I have however two points that are not very well covered in my opinion in the manuscript.

First, the authors provide very little assessment of the error in the ancestral state reconstruction. It is well known that the deeper you go in the phylogenetic tree, the higher is the uncertainty in the ancestral state estimation. This is true for both maximum likelihood and bayesian approaches. However, the results are not showing this at all (e.g. Fig 1 or 3, for instance). The extended data figure 1, that is suppose to represent this is quite unclear in my opinion and the full text is written as if the basal flowers were estimated without any uncertainty. This should be modified absolutely to better inform the reader about the reliability of your results.

RESPONSE: We thank you for making these comments and absolutely agree that the reader should be informed about the relative reliability of our results (some are confident,

others remain more uncertain). In the original version, most uncertainty was reported in full detail and discussed in the supplementary information (especially Supplementary Table 2 and the Supplementary Discussion), as noted by Reviewer #3. Uncertainty was also reported very explicitly, as 95% credibility intervals, in Fig. 1. However, as we explain below, we have now thoroughly revised the main text to leave no ambiguity in this respect.

We feel very strongly about this important aspect of our study, as we believe this is one of very few studies that have sought to explore and quantify uncertainty to the levels presented here: we submitted our dataset to three methods of ancestral state reconstruction, five different sets of higher-level relationships in angiosperms (each represented by >1000 phylogenetic trees), two contrasting hypotheses for the age of angiosperms, and many models of morphological evolution. We think this is an important original contribution of our approach that will be interesting to many readers outside the botanical community.

Specifically, we have made the following modifications:

1) We added a statement on uncertainty in the Abstract:

“Although uncertainty remains for some of the characters, our reconstruction uncovers a new plausible scenario for the early diversification of flowers, leading to new testable hypotheses for future research on angiosperms.” (Abstract, lines 12-15)

2) We made a more explicit reference to the extensive Supplementary Information at the beginning of the Results section where we first report on the estimated morphology of the ancestral flower:

“We infer that the flower of the most recent common ancestor of all living angiosperms (hereafter referred to as the ancestral flower) was most likely bisexual and had an undifferentiated perianth of more than ten tepals, an androecium of more than ten stamens, and a gynoecium of more than five carpels. We also infer that the perianth and the androecium probably had whorled phyllotaxis with three organs per whorl. Taken together, these numbers imply at least four whorls in each organ category (Fig. 1; see Supplementary Table 2 and Supplementary Discussion for estimates of uncertainty associated with ancestral states).” (Results, lines 66-74)

3) We added an entire new section (two paragraphs) in the Results. Note that the first paragraph is a modified version of a short introduction that we had placed (and still is) in the Supplementary Discussion. The second paragraph summarizes the main points that were made (and still are) on this topic in the Supplementary Discussion, but also addresses your comment on the relationship between node depth and uncertainty.

“Uncertainty in ancestral state reconstructions. *Estimating features of the ancestral flower is a difficult task, because there are neither suitable outgroups for direct comparison^{5,15}, nor fossil flowers known from the time period when this ancestor existed³¹. In this study, we made these inferences based on the distribution of traits in extant angiosperms and their phylogenetic relationships, and, for the first time, methods using explicit models of stochastic evolution for morphological characters. While these analyses help us resolve long-standing ambiguities (e.g., ancestral sex) and reconstruct ancestral flowers at internal key nodes rarely assessed in previous work (e.g., Pentapetalae), such reconstructions necessarily come with limitations and some uncertainty. Ignoring uncertainty would entail the risk of presenting our reconstruction as definitive, which may be an impossible goal.*

Through our detailed comparison of three reconstruction methods, five series of trees (each sampling >1000 chronograms), two timescales for the angiosperms, and many models of morphological evolution, we found that reversible-jump Bayesian methods performed best at measuring uncertainty in ancestral state reconstruction, whereas maximum likelihood nearly always suggested misleadingly high confidence (Supplementary Discussion). For this reason, 95% credibility intervals obtained from the reversible-jump Bayesian analyses are reported throughout this study (Fig. 1; Supplementary Table 2). This is an important step forward because previous higher-level studies of floral evolution focused almost exclusively on parsimony reconstructions and lacked any assessment of uncertainty associated with ancestral states. Furthermore, early work on ancestral state reconstruction suggested a positive relationship between uncertainty and node depth³², which would predict that all ancestral states reconstructed for the root of our angiosperm tree should be uncertain. Interestingly, we found that this is not always true (about half of the floral traits examined yielded highly confident estimates; Extended Data Fig. 1; Supplementary Discussion), although we observed that focal nodes nested in Monocotyledoneae and Eudicotyledoneae were on average reconstructed with higher confidence than deeper nodes.” (Results, lines 91-119)

4) We added new sentences at the beginning of the next section (“A new scenario for the early evolution of flowers”) to clarify that ancestral floral phyllotaxis is uncertain (a topic we had made very explicit and discussed extensively in the Supplementary Discussion), yet this uncertainty in itself challenges the traditional dogma on this question:

“Although reconstruction of ancestral floral phyllotaxis proved relatively uncertain in this study (Supplementary Discussion), as in previous work based on parsimony alone⁶⁻⁸, the implications of our result are important to consider for two reasons. First, the idea that whorled phyllotaxis of floral organs always evolved from spiral phyllotaxis is still prevalent among botanists. Our analyses provide the most comprehensive evidence so far that the opposite is more likely.” (Results, lines 123-128)

5) We have inserted statements on uncertainty in the legends of Figures 1 and 3. This is useful because some people (incl. teachers, journalists) might be tempted to present these figures as single, definitive answers, which we hope we have made clear is not the case in our revised text and extensive Supplementary Information.

“Bold states indicate high confidence and consistency across methods of reconstruction (e.g., perianth present, undifferentiated, and actinomorphic). Other states need to be interpreted with caution as their reconstruction was either associated with higher uncertainty (e.g., perianth phyllotaxis, number of stamen whorls) or inconsistent across methods (e.g., sex reconstructed as equivocal with parsimony).” (Figure 1 legend, lines 671-676)

“Note that there is uncertainty associated with some of these reconstructions (especially for Angiospermae, Magnoliidae, and Eudicotyledoneae). Therefore, the scenario illustrated here is one of several plausible alternatives and should be taken with caution.” (Figure 3 legend, lines 698-701)

6) We have modified Extended Data Fig. 1 to clarify what it represents by adding X and Y labels and an inset legend for the colour codes.

7) Last, we have opted to keep the numbers out of the main text (i.e., refer to the Supplementary Information for them) in order to maintain an easily accessible streamlined article. However, we will be very happy to move the numbers (rjMCMC CIs) into the main text should you think it necessary.

Collectively, we believe these clarifications do not diminish the significance of our results, and instead contribute to an improved presentation of our study.

Second, the ancestral state reconstructions are done for every morphological characters independently. This can be justified from a methodological point of view as it is easier to handle the reconstruction for each character one by one, but it makes little sense biologically as the evolution of the different parts of the flower are certainly linked and evolve under some correlation. I fully admit that it is not easy to account for such a dependency between traits, but there are ways to at least assess the level of correlation and discuss it in the manuscript. The authors should look at the papers by Adams and Felice (2014) or Bartoszek et al (2012) for a discussion and some possible ideas on how to do this.

RESPONSE: We fully agree and are happy to present an entirely new set of correlation analyses in response to these comments, as explained below. Note that, unfortunately, the very interesting multivariate solutions proposed by Adams and Felice (2014) and Bartoszek et al. (2012) were designed for continuous characters and thus cannot be applied to the discrete characters analyzed in this study. We have made this clear in the new text (see below).

Briefly, we analyzed every possible pair of floral traits in our dataset, after removing duplicates (multiple versions of the same character) and reducing all multistate characters to binary versions of the same character. We had to do this because current methods for analyzing discrete character correlations, grounded on Pagel's (1994)'s models and ideas, only allow us to test pairwise correlations among binary traits. A multivariate framework, jointly analyzing the correlation among many characters, would be a more desirable approach but is currently lacking for discrete character evolution, partly because of the gigantic number of parameters and possible models that would be involved. O'Meara et al. (2016) recently proposed a particularly innovative solution to handle the problem, but this is based on an oversimplification of model space that we do not feel comfortable with and, more importantly, the analyses involved would be computationally very demanding (especially for our dataset, which includes many more traits than the six traits they considered) and thus were not realistic to implement for this paper. Therefore, we acknowledge that the solution we propose is not perfect, but still allows us to move forward in assessing the plausibility of our results. In total, we analyzed all possible 231 pairwise correlations among the 22 traits that remained after the modifications presented above. Each correlation was analyzed with both maximum likelihood and reversible-jump MCMC approaches very similar to those we used for single-trait analyses. Because these new analyses are computationally demanding, we focus on one particular tree series (the 'C' set), though we expect results would be similar for the other tree series as they were for single-trait analyses.

Our results are quite interesting. We found a level of significant correlations higher than we would have expected based on developmental or functional constraints. Indeed, 40-48% of the pairs tested appear to be correlated. However, close examination of the ancestral combined states reconstructed from these analyses revealed results generally consistent with the main single-trait analyses reported in the paper.

To present and discuss these new results, we have added a new table (Table 1, the pairwise correlation matrix) and supplementary table (Supplementary Table 3, providing the complete set of results in a format similar to Supplementary Table 2).

In addition, we briefly mention the results in a new short paragraph added to the main text:

“We also evaluated the level of correlation among floral traits and its impact on reconstructed ancestral states. We found significant support for correlated evolution in 40-48% of the pairs tested (Table 1), a result consistent with previous studies of floral integration²⁵⁻³⁰. However, accounting for these correlations did not substantially affect the results obtained from analyses of individual traits (Supplementary Table 3; Supplementary Discussion).” (Results, lines 85-90)

Then, we also added an entire new section in the Methods to describe how we conducted these analyses:

“Correlation analyses. *As flowers are highly complex and integrated structures, floral traits are unlikely to evolve independently from one another²⁵⁻³⁰. Although our main goal was not to evaluate the level of morphological integration in flowers, it is possible that such correlations might impact ancestral state reconstructions. However, in contrast to recently developed multivariate approaches for continuous characters⁶⁷⁻⁶⁹, no comparative method exists yet to account for the potential correlation of more than two discrete characters, unless a drastic simplification of model space is made²⁵. In addition, correlated models and analyses have typically been developed for binary characters^{57,60}. The stochastic mapping approach to correlation tests allows inclusion of multistate characters, but does not model character correlation and starts at the onset by reconstructing ancestral states independently at all nodes⁷⁰; it was thus not relevant to our specific objective here. Therefore, we tested correlations among all possible pairs of binary floral traits in our dataset. To do so, we first removed redundancies for multiple versions of the same character (Supplementary Methods), and then transformed all multistate characters into binary characters by maintaining the hypothesized ancestral state for the angiosperms as one state and pooling the remaining states as another (e.g., for the number of perianth whorls, we analyzed one-two whorls vs. more than two whorls). We thus obtained a new set of 22 presumably independent characters and analyzed all 231 pairwise correlations among these characters (Table 1). Given our observation that reconstructed ancestral states in the single-trait analyses were remarkably consistent across the 10 series of phylogenetic trees (see Supplementary Discussion), we conducted all of our correlation analyses using the C series of trees, which best reflects the current consensus on higher-level angiosperm phylogeny and allows us to take into account phylogenetic uncertainty.*

As for our single-trait analyses, we used both a maximum likelihood (ML) and a Bayesian reversible-jump MCMC (rjMCMC) approach to test for correlations and their impact on reconstructed ancestral states, using again the rayDISC function of corHMM 1.18⁵⁵ in R⁵⁴ for ML analyses and BayesTraits 2⁵⁸ for rjMCMC analyses. The ML approach allowed us to test the fit of a small set of combined Markov models (i.e., with 4x4 Q matrices to model all possible transitions among the four possible combined states, excluding dual transitions), including correlated (dependent) and uncorrelated (independent) models⁶⁰. Specifically, for each character pair, we fitted four correlated models (ARDnodual, ARDnodualeq, differing only in the root state prior: see above; SYMnodual, SYMnodualeq) and three uncorrelated models (ERNodual, UNCORRnodual, UNCORRnodualeq; UNCORRnodual corresponds to the most general, 4-parameter “independent” model of ref⁶⁷). Using AICc, we selected the best-fit model and compared the ancestral combined states reconstructed with those obtained in our single-trait analyses (Supplementary Table 3). As a measure of support for correlation, we report the cumulative Akaike weight of correlated models (Table 1). The rjMCMC approach allowed us to explore the vast space of the 21,146 possible Markov combined models for the evolution of two binary characters, sampling models according to

their posterior probability⁵⁷, with settings as above (10M generations, sampling every 100 generations). As for single-trait analyses, the ancestral states reconstructed using this approach integrate over model, parameter, tree, and dating uncertainty, as measured by the credibility intervals associated with the probability (proportional likelihood) of each state (Supplementary Table 3). Support for correlation is here measured by the Bayes Factor comparing the dependent models to the independent models, rewritten as the ratio of the posterior to the prior odds of the two models⁵⁷: $BF_{DI} = [P(M_D|D)/P(M_I|D)] / [(21146 - 51)/51]$, where $P(M_D|D)$ and $P(M_I|D)$ are the sampling frequencies of dependent and independent models, respectively.” (Methods, lines 401-452)

Last, we present and discuss these results in more detail in an entire new section of the Supplementary Discussion (“Results from the correlation analyses”; see Supplementary Discussion, pages 32-35).

In summary, we think this was a very useful and interesting complementary addition to our study of floral traits. We had initially planned to conduct such analyses in a follow-up paper, but we find that the correlation results now presented in this paper actually represent an excellent starting point for future work on floral integration at the angiosperm level. Thus, we expect that these new results, too, will generate much interest.

Reviewer #3 (Remarks to the Author):

The authors attempt to reconstruct the ancestral flower using the distribution of floral traits among extant angiosperms. They assemble a dataset of traits from the literature and apply model-based methods to reconstruct ancestral states on alternative phylogenetic trees for angiosperms. They infer that the ancestral flower was most likely bisexual and actinomorphic, with an undifferentiated perianth of four trimerous whorls of free tepals, an androecium of four trimerous whorls of introrse stamens, and a gynoecium of at least six free, superior carpels. This exact combination of characters does not exist in extant angiosperms, and the authors offer scenarios of early floral evolution to explain the floral structure of derived angiosperm clades.

The authors acknowledge that the task of reconstructing the ancestral flower is a difficult one, given the lack of suitable fossils and outgroups, and the variability of floral structure in the basal grade (ANA grade) of angiosperms. Therefore, it is anticipated that they would provide compelling evidence should they arrive at robust conclusions where previous studies have been ambiguous. Disappointingly, the Main Text and supplementary information do not meet these expectations, and are often in conflict in with each other.

RESPONSE: Thank you for taking time to critically examine our manuscript, including your careful scrutiny of our Supplementary Information. Below we reply to the two important comments you made here.

1) As outlined above in our response to Reviewer #2, we have now made it clearer that not all of our results are equally robust. We cannot provide compelling evidence for our less robust results, and might never be able to. Despite this, and even if we do reach less confident answers for some long-standing parsimony ambiguities, we are still convinced that

a thorough, model-based examination of the specific question of the ancestral flower, taking into account many variable parameters, was necessary and represents a very significant step forward into elucidating the early diversification of flowers. We also would like to emphasize that some form of uncertainty is expected for what we attempt to infer about a flower that existed at least 140 millions of years ago and produced the extraordinary diversity of flowers observed in extant species — finding no uncertainty would be suspicious. Part of the originality of this study, beyond the size of the dataset and the model-based methods, also lies in our extensive efforts to quantify uncertainty associated with ancestral state reconstructions, whereas most previous work reported either one most parsimonious state or two equivocal ones, without any measure of error.

Now, we would also like to point out that not all of our results are uncertain. A key example that is already well emphasized in the text and illustrated in Figure 2 is the compelling evidence we provide with model-based approaches that the ancestral flower was bisexual, whereas parsimony analyses, past and present, leave this equivocal due to the functionally unisexual flowers of *Amborella*. This is a major achievement. It has been a key question for a very long time, with direct implications for understanding the origin of the flower. Although very recent papers have argued that indeed, the ancestral flower is more likely to have been bisexual (e.g., Endress & Doyle 2015; Anger et al. 2017), this study is, to our knowledge, the first one ever to present actual support for this view.

Further, our study is not restricted to the question of the ancestral flower of angiosperms as a whole or at the most basal nodes within their phylogeny. Most previous work, including that of two authors of this paper (James Doyle and Peter Endress), had focused on these nodes as well as Magnoliidae. This study instead was designed to sample all angiosperm orders and most families and thus allowed us to reconstruct ancestral states in all of the internal nodes of the backbone of angiosperm phylogeny. In addition to Monocotyledoneae and Eudicotyledoneae, which have been touched upon in previous studies, this includes Commelinidae, Pentapetalae, Asteridae, Rosidae, etc. The ancestral flowers for many of these key clades (containing most of extant species of angiosperms) had never been reconstructed, let alone addressed with model-based approaches and a rich taxon sample. In addition, most (though not all) of these ancestral flowers are reconstructed with greater confidence than that of angiosperms, and we provide compelling evidence for the robust results we find for these largely unexplored questions. For instance, the pentamery of Pentapetalae, an absolutely essential result for moving forward in understanding floral evolution in eudicots, though extensively discussed in some papers, had only been tested so far (to our knowledge) by Ronse de Craene (2008), using a phylogeny of angiosperms known at that time and parsimony optimization. He found the ancestral state for this character to be equivocal at this node, whereas all of our analyses (parsimony, ML, rjMCMC) across all 10 tree series reconstructed pentamery as ancestral, with very strong support.

2) We understand that the main text and Supplementary Information may have appeared to be in contradiction regarding the issue of uncertainty. As explained in detail in our response to Reviewer #2, who also pointed to the same problem, we have now extensively revised the main text (and figure legends) to make it clear that not all results are equally certain (as one might expect) and avoid any misleading impression of readers in that direction. These revisions have also allowed us to bring out the originality of this study in capturing a reliable measure of uncertainty (see response above, in particular the new Results section named “Uncertainty in ancestral state reconstructions”). We thank you for bringing this important point to our attention and hope that you will find our revision satisfactory.

Last, we would kindly like to object your suggestion that there was ‘conflict’ between the two parts of the manuscript, as we explain below for the examples you pointed out. However, we are grateful for this criticism as it allowed us to clarify some wordings that were apparently misleading.

Specific examples of conflict include:

1. Ancestral perianth phyllotaxis:

Main text: “Our study provides the first tentative evidence that the ancestral flower of all angiosperms had a perianth (tepals) and an androecium (stamens) organized in whorls, rather than in a spiral.”

Supplemental Discussion: “Our analyses also suggest that perianth phyllotaxis (char. 230_A) of the ancestral flower was whorled, although we acknowledge that support for this result remained low.”

The low support should be mentioned in the main text. Moreover, whether their choice to “... score perianth phyllotaxis only at anthesis because developmental data are lacking for most species in our data set.”, affected their character reconstruction should also be explored.

RESPONSE: There is no true conflict here and the relatively low support was already implied in the term “tentative”, but we agree this could easily be missed. We have added “most likely” in this sentence to further convey the idea that this result is not certain and we elaborate on this uncertainty (and why we think the lack of evidence for a spiral ancestor is important in itself) in the following sentences:

“Our study provides the first tentative evidence that the ancestral flower of all angiosperms most likely had a perianth (tepals) and an androecium (stamens) organized in whorls, rather than in a spiral. Although reconstruction of ancestral floral phyllotaxis proved relatively uncertain in this study (Supplementary Discussion), as in previous work based on parsimony alone⁶⁻⁸, the implications of our result are important to consider for two reasons. First, the idea that whorled phyllotaxis always evolved from spiral phyllotaxis is still prevalent among botanists. Our analyses provide the most comprehensive evidence so far that the opposite is more likely. Second, this result, if correct, would imply [...]” (Results, lines 120-129)

The question of phyllotaxis at the onset of development is difficult for two reasons. First, as the Reviewer noted, we explain that reliable data on floral development are lacking for most species in our dataset, therefore it is not possible to explore, at this stage, what the results would be for the optimization of a developmental character (without excessive speculation). Second, the question is complicated by the fact that successive temporal initiation of primordia in a spiral-like pattern does not necessarily imply a spiral phyllotactic spatial pattern, which can only be determined from divergence angles. The literature is confusing in this respect, and good data, illustrations, and descriptions that would allow us to distinguish between temporal and spatial arrangements are very few. For these reasons, and because we have already made our scoring choice very explicit in the Supplementary Methods (as noted by the Reviewer), we prefer not to elaborate on this point in the main text.

2. Ancestral number of perianth organs:

Main text, line 69 page 4: “We infer that the ancestral flower was most likely bisexual and had an undifferentiated perianth of four whorls of three tepals each...”

Supplemental Discussion, page 26: “... we have decided to tentatively portray the reconstructed ancestral perianth with four whorls of three parts each (12 in total; Fig. 1, Fig. S2), but we acknowledge that other alternatives with fewer or more parts cannot be ruled out based on our analyses.”

Some mention of the uncertainty surrounding the number of perianth organs seems warranted in the main text. Also, and importantly, it is very unclear how the authors arrived at four for the number of whorls of tepals. I cannot find a justification for this number in the main text, and the supplemental discussion seems to suggest that more than two whorls was their most precise reconstruction:

Pg 25: “...we consider it more likely that the ancestral flower had a perianth formed of more than two whorls.”

Pg 26: “we cannot reconstruct the ancestral number of perianth whorls more precisely.”

Possibly, the number four was reached because the total number of perianth parts was reconstructed as “more than 10” in ML and rjMCMC analyses (Supplemental Discussion, page 25) and therefore the fewest trimerous whorls required to accommodate more than 10 tepals is 4. If so, this conclusion seems somewhat arbitrary to me.

RESPONSE: Thank you for pointing this out. There is, however, no true conflict here either. Because we inferred that both the perianth and androecium each had more than ten parts and, if whorled, both were trimerous and had more than two whorls, the only intersection of these traits is a perianth and an androecium each of at least four trimerous whorls. You are correct that, for the purpose of graphical representations, we chose to conservatively depict the minimum possible number of whorls (four). Because this condition is very rare in angiosperms and nearly no extant species have five or more whorls, it may never be possible to reconstruct the exact number of whorls based on ancestral state reconstructions only, without complementary fossil or evo-devo evidence (currently lacking).

We thought this was clear in our Supplementary Discussion, but this is indeed very important and thus we have now clarified the inference by expanding the relevant section of the main text:

“We infer that the flower of the most recent common ancestor of all living angiosperms (hereafter referred to as the ancestral flower) was most likely bisexual and had an undifferentiated perianth of more than ten tepals, an androecium of more than ten stamens, and a gynoecium of more than five carpels. We also infer that the perianth and the androecium probably had whorled phyllotaxis with three organs per whorl. Taken together, these numbers imply at least four whorls in each organ category (Fig. 1; see Supplementary Table 2 and Supplementary Discussion for estimates of uncertainty associated with ancestral states).” (Results, lines 66-74)

Please note that we also updated the Abstract accordingly:

“We reconstruct the ancestral angiosperm flower as bisexual and radially symmetric, with more than two whorls of three separate perianth organs each (undifferentiated tepals), more than two whorls of three separate stamens each, and more than five spirally arranged separate carpels.” (Abstract, lines 8-12)

To remove any possible ambiguity or risk of misinterpretation, we also clarified this point in the legend of Figure 1:

“The exact number of organs could not be reconstructed precisely. Minimum numbers were chosen for this representation, but reconstructions with more floral organs are also compatible with our results (see Supplementary Information for further details).” (Figure 1 legend, lines 678-681)

3. Ancestral number of stamens:

Main text, line 69 page 4: “We infer that the ancestral flower was most likely bisexual and had, an androecium of four whorls of three stamens each...”

Supplemental Discussion, page 26: we tentatively chose to depict the ancestral androecium as formed of four whorls of three parts each (as the perianth), but we acknowledge that other alternatives are possible.

Same comments as for perianth numbers above.

RESPONSE: See our response above, which deals with this question for both the perianth and the androecium.

4. Ancestral number of carpels:

Main text, line 69 page 4: “We infer that the ancestral flower was most likely bisexual and had, and a gynoecium of at least six carpels...”

Supplemental Discussion, page 27: “The ancestral flower most likely had a superior ovary ... with more than five carpels...”

Similar to the numbers for perianth organs and stamens, it is not entirely clear how the authors reconstruct exactly six carpels in the main text while the modeled character state was “more than five carpels”.

RESPONSE: Yes, this was confusing, thank you for pointing it out to us. Although “at least six” and “more than five” are exactly the same thing, we certainly admit that there was no need for two wordings. As for the number of perianth and androecium whorls, six carpels is also a conservative minimum number chosen for the sake of graphical representation. We have now modified the Abstract, Results, and legend of Figure 1 to make the wording “more than five” consistent throughout and clarify that the exact number of carpels is not known (see quotes above).

Thus, for all four of these features of the ancestral flower, the manuscript is confusing or unconvincing. Other ancestral features discussed are not controversial and generally support the analyses of earlier publications (e.g, Endress and Doyle 2015).

RESPONSE: We hope we have answered your concerns in this revised version. With respect to the question of the novelty of our approach, we refer to the assessments made by the other two reviewers. In addition, as we have explained briefly above, our study presents many new findings for nodes and traits rarely evaluated before and provides the long-needed model-based, complementary answer to the parsimony studies published so far and, for the first time, a thorough assessment of the uncertainty associated with reconstructed ancestral states. This is very timely, as a large proportion of the macroevolutionary community now routinely employs probabilistic methods to address similar questions in other clades, at different scales, or for other traits. We do not contend that some of our new findings are not controversial, but we firmly believe that thorough tests and analyses such as those presented in this study are the very useful, and often badly needed complement to hand-waving ideas and discussions still prevalent in some of the literature on floral evolution. Only with large datasets and analyses can we move forward in answering many questions in the fascinating evolutionary diversification of flowers and we hope this study will represent a major new landmark in this direction.

REVIEWERS' COMMENTS:

Reviewer #1 (Remarks to the Author):

The authors have satisfactorily responded to my concerns. I've made a few minor comments directly on the MS.

Reviewer #2 (Remarks to the Author):

The revised manuscript by Sauquet et al incorporated most of the comments made by the different reviewers in the previous round of reviews. I am pleased to see that my specific comments on the uncertainty of the ancestral reconstructions and the correlation between characters were considered. The modifications are improving the manuscript and I have no other comments to make at this point.

Reviewer #3 (Remarks to the Author):

In reading this revised manuscript, I find that the authors have satisfactorily addressed the concerns I raised about the original manuscript.

I am happy to leave the final decision to the editors.

REVIEWERS' COMMENTS:

Reviewer #1 (Remarks to the Author):

The authors have satisfactorily responded to my concerns. I've made a few minor comments directly on the MS.

RESPONSE: Thank you for your assessment of our revised manuscript. We are happy that you found the changes satisfactory and we thank you for helping us improve this paper. Below we have pasted the additional minor comments you made in a PDF copy of the manuscript and respond to each of them.

Line 98: 'sex of angiosperm flowers' (sexual system/plant gender are broader, so make clear here what focus is?)

RESPONSE: We have clarified this by rewording the parenthetical statement:

"While these analyses help us resolve long-standing ambiguities (e.g., whether the ancestral flower was bisexual or unisexual) [...]" (Results, lines 98-100)

Line 101: This seems unduly negative (and perhaps wrong if future fossil information is uncovered). You could instead say that the associated uncertainty is quantifiable?

RESPONSE: We have deleted this sentence and replaced it with this new one, following your suggestion:

"However, it should be possible to quantify this uncertainty." (Results, line 104)

Line 103: give a brief idea here of what the chronograms were derived from?

RESPONSE: We have now provided this information:

"Through our detailed comparison of three reconstruction methods, five series of trees (each sampling >1000 chronograms obtained from fossil-calibrated divergence time analyses in BEAST), [...]" (Results, lines 105-107)

Line 128: unless of course an earlier stem angiosperm had spiral phyllotaxis!

RESPONSE: We agree and have clarified this important point by expanding this sentence:

“Our analyses provide the most comprehensive evidence so far that the opposite is more likely within crown-group angiosperms (this does not preclude the possibility that the ancestral flower was itself derived from a spiral ancestor further down the stem lineage of the group).” (Results, lines 131-134)

Line 143: Both Liliaceae and Arabidopsis have 'outermost perianth whorls' by definition. Can you rephrase (e.g. with reference to loss from an ancestor) to make this a bit clearer?

RESPONSE: We have clarified this by inserting 'ancestral two' before 'outermost perianth whorls'. We think this is enough here and would prefer to leave the detailed explanation of this possibility in the Supplementary Discussion as it is now). However, we realized that it was not clear to all co-authors that this was just one of several alternatives. Therefore, we added a second sentence to clarify this:

“[...] suggesting that the ‘sliding boundary’ ABCE model of Liliaceae could in fact be a conserved Arabidopsis ABCE model expressed in reduced flowers lacking the ancestral two outermost perianth whorls. However, other alternatives exist, including one where the two perianth whorls of Monocotyledoneae are homologous with the calyx (outer perianth whorl) of Pentapetalae by loss of the ancestral two innermost perianth whorls.” (Results, lines 148-153)

Line 150: I'd phrase as 'divergence and canalization' (in that order) as canalization is presumably usually secondary, to reduce sensitivity of trait expression to environmental variation?

RESPONSE: Thank you for pointing this out. We have now modified the sentence following your suggestion:

“Second, it is possible that a reduced number of perianth whorls facilitated the divergence and canalization of genetic programs among whorls, leading to the strong perianth differentiation into sepals and petals that is characteristic of most members of Pentapetalae¹⁶.” (Results, lines 159-163)

Line 230: Next para should start here I think

RESPONSE: We prefer not to make this change, because this sentence (starting with “Integrating”) is the logical consequence of the limitations expressed in the three previous sentences. To clarify this flow, we have now added “Thus” at the beginning of this sentence:

“Thus, integrating phylogenetic uncertainty in our Bayesian analyses of trait evolution was the primary motivation for re-analyzing the dataset in BEAST without fixing the topology.” (Methods, lines 241-243)

Line 244: I think you mean 'a clade comprising M+C+E' - otherwise it is ambiguous what is monophyletic

RESPONSE: This is correct. To remove the ambiguity, we have now rephrased this:

“The C series of analyses refers to the same set-up as the B series, but with two topological constraints for deep-level angiosperm relationships: (1) Amborella sister to the rest of angiosperms; (2) Monocotyledoneae + Ceratophyllaceae + Eudicotyledoneae together forming a clade (excluding Chloranthaceae and Magnoliidae; Supplementary Fig. 1).”
(Methods, lines 253-257)

Reviewer #2 (Remarks to the Author):

The revised manuscript by Sauquet et al incorporated most of the comments made by the different reviewers in the previous round of reviews. I am pleased to see that my specific comments on the uncertainty of the ancestral reconstructions and the correlation between characters were considered. The modifications are improving the manuscript and I have no other comments to make at this point.

RESPONSE: Thank you for your assessment of our revised manuscript. We are happy that you found the changes satisfactory and we thank you for helping us improve this paper.

Reviewer #3 (Remarks to the Author):

In reading this revised manuscript, I find that the authors have satisfactorily addressed the concerns I raised about the original manuscript.

I am happy to leave the final decision to the editors.

RESPONSE: Thank you for your assessment of our revised manuscript. We are happy that you found the changes satisfactory and we thank you for helping us improve this paper.